# DATA ATTRIBUTION FOR MULTITASK LEARNING

## ABSTRACT

Data attribution quantifies the influence of individual training data points on machine learning models, aiding in their interpretation and improvement. While prior work has primarily focused on single-task learning (STL), this work extends data attribution to multitask learning (MTL). Data attribution in MTL presents new opportunities for interpreting and improving MTL models while also introducing unique technical challenges. On the opportunity side, data attribution in MTL offers a natural way to efficiently measure task relatedness, a key factor that impacts the effectiveness of MTL. However, the shared and task-specific parameters in MTL models present challenges that require specialized data attribution methods. In this paper, we propose the *MultiTask Influence Function* (**MTIF**), a data attribution framework tailored for MTL. MTIF leverages the parameter structure of MTL models to derive influence functions that distinguish between within-task and cross-task influences. Our derivation also sheds light on the applicability of popular approximation techniques for influence function computation, such as EK-FAC and LiSSA, in the MTL setting. Compared to conventional task relatedness measurements, MTIF provides not only task-level relatedness but also data-level influence analysis. The latter enables fine-grained interpretations of task relatedness and facilitates a data selection strategy to effectively mitigate negative transfer in MTL. Extensive experiments on both linear and neural network models show that MTIF effectively approximates leave-one-out and leave-one-task-out effects while offering interpretable insights into task relatedness. Moreover, the data selection strategy enabled by MTIF consistently improves model performance in MTL. Our work establishes a novel connection between data attribution and MTL, offering an efficient and scalable solution for measuring task relatedness and enhancing MTL models.

## 1 INTRODUCTION

Data attribution aims to quantify the influence of individual training data points on machine learning models and has been widely used to interpret and improve these models (Koh & Liang, 2017; Hammoudeh & Lowd, 2024). However, most existing literature on data attribution focuses on single-task learning (STL) settings. In contrast, this work explores data attribution in the context of multitask learning (MTL), where multiple related tasks are trained simultaneously to enhance overall performance (Caruana, 1997). Data attribution in MTL presents new opportunities for interpreting and improving MTL, with distinct technical challenges in comparison to data attribution in STL.

MTL has demonstrated success across a wide range of domains, including computer vision (Zamir et al., 2018), natural language processing (Hashimoto et al., 2017), speech processing (Huang et al., 2015), and recommender systems (Ma et al., 2018). In practice, however, MTL does not always help with the overall performance—training unrelated tasks together often harms the learning performance, a phenomenon known as negative transfer (Standley et al., 2020; Wang et al., 2020; Parisotto et al., 2016; Rusu et al., 2016). As a result, understanding and quantifying task relatedness has become a key focus in MTL research (Ma et al., 2018; Standley et al., 2020; Fifty et al., 2021). Despite this, there is still no consensus on a universally effective and efficient method for measuring task relatedness. In practical applications, practitioners often rely on trial and error—repeatedly training models with different task combinations—as a gold standard to assess task relatedness, a process that is computationally expensive.

Generalizing data attribution methods to MTL offers a promising, efficient, and interpretable way to measure task relatedness in MTL. Many data attribution methods are designed to efficiently approx-

imate the change of model performance when retraining the model with certain data points excluded from the training dataset (Koh & Liang, 2017; Park et al., 2023). Extending these methods to MTL naturally leads to an efficient approximation of the aforementioned trial-and-error process for determining task relatedness. Moreover, data attribution methods allow for fine-grained analysis at the individual data instance level, revealing how data points from one task impact performance on another task. This data-level influence analysis offers more interpretable insights into task relatedness by moving beyond a single metric, providing concrete evidence of how tasks are related through specific data points and their cross-task effects.

However, MTL introduces unique challenges that require tailored data attribution methods. MTL models typically consist of both shared parameters across all tasks and task-specific parameters for each individual task. When making predictions for a specific task, only a submodel with a subset of the parameters is utilized. As the number of tasks increases, this brings several computational challenges for data attribution. Firstly, since each task corresponds to a separate attribution target, retraining-based data attribution methods (Ghorbani & Zou, 2019; Jia et al., 2019) become prohibitively expensive. Therefore, in this paper, we focus on influence function (IF)-based data attribution methods that do not require repeated retraining. Additionally, tasks in MTL may employ different loss functions, and the number of parameters scales with the number of tasks, further complicating the application of existing IF-based data attribution methods designed for STL. These factors present significant technical challenges when adapting such methods to the MTL setting.

In this paper, we propose the *MultiTask Influence Function* (**MTIF**) to address these challenges. Similar to the IF-based data attribution methods for STL (Koh & Liang, 2017), MTIF leverages a first-order approximation to efficiently estimate the impact of removing a data point from one task on the prediction for another task, without the need for model retraining. Specifically designed for MTL, MTIF derives the influence of data points on the shared and task-specific parameters separately, and exploits the unique structure of MTL models to enhance computational efficiency. MTIF enables the efficient estimation of both data-level and task-level influence, providing a scalable and interpretable solution for data attribution in MTL settings.

We conduct extensive experiments on both linear and neural network models to evaluate the effectiveness of the proposed MTIF. On linear models, the data-level influence scores predicted by MTIF shows a near perfect correlation with the actual change of model outputs obtained by brute-force leave-one-out retraining; the task-level influence estimated by MTIF also strongly correlates with the leave-one-task-out retraining, with an average Pearson correlation around 0.7. On neural network models, the task-level influence estimated by MTIF also shows significant correlation with leave-one-task-out retraining, with Pearson correlation ranging from 0.1 to 0.4. Moreover, the data-level influence estimated by MTIF enables fine-grained data selection for MTL, which demonstrates consistent performance improvements over baselines. Finally, we provide case studies of the most negative data points from one task to another task, providing interpretations about negative transfer.

We summarize the contributions of this study as following. Firstly, we establish a novel connection between data attribution and MTL, where the former can be naturally employed to efficiently measure task relatedness, a key concept in MTL. Secondly, we propose a scalable data attribution method, MTIF, that addresses unique challenges of data attribution in MTL and provides fine-grained influence analysis. Finally, we demonstrate the effectiveness and practical usefulness of the proposed method through extensive experiments.

## 2 RELATED WORK

**Data Attribution** Data attribution methods quantify the influence of individual training data points on model performance. These methods can be broadly categorized into retraining-based and gradient-based approaches (Hammoudeh & Lowd, 2024). Retraining-based methods (Ghorbani & Zou, 2019; Jia et al., 2019; Kwon & Zou, 2022; Wang & Jia, 2023; Ilyas et al., 2022) require retraining the model multiple times on different subsets of the training data. Retraining-based methods are usually computationally expensive due to the repeated retraining. The computation cost can be further exacerbated in MTL due to the combination of tasks. Gradient-based methods (Koh & Liang, 2017; Guo et al., 2021; Barshan et al., 2020; Schioppa et al., 2022; Kwon et al., 2024; Yeh et al., 2018; Pruthi et al., 2020; Park et al., 2023) instead rely on the (higher-order) gradient information of the original model to estimate the data influence, which are more efficient. Many gradient-based

methods can be viewed as variants of IF-based data attribution methods (Koh & Liang, 2017). In this paper, we develop an IF-based data attribution method tailored for the MTL settings.

**Task Relatedness in Multitask Learning** Quantifying task relatedness has been a central focus in multitask learning. Broadly, two lines of work address this topic. The first focuses on task grouping or task selection, aiming to develop methods for grouping or selecting positively related tasks to improve prediction performance. Standley et al. (2020) and Li et al. (2023) introduced task selection methods based on model retraining, which are less efficient than our method. Fifty et al. (2021) proposed an efficient method for calculating heuristic pairwise task affinities, but their estimator heavily depends on the training trajectory, which limits its interpretability. Additionally, Wu et al. (2020) incorporated task data covariance to estimate task similarity, though their work is restricted to specific types of models. Another line of research focuses on developing advanced training algorithms for MTL by explicitly accounting for inter-task relations during training. These methods generally fall into two categories. The first category manipulates per-task gradients to mitigate negative influences between tasks (Yu et al., 2020; Wang et al., 2021; Liu et al., 2021a; Chen et al., 2020; Peng et al., 2024). The second category employs task reweighting techniques to balance the contribution of each task or to emphasize on critical tasks (Chen et al., 2018; Liu et al., 2019; Guo et al., 2018; Kendall et al., 2018). Beyond these two categories, Duan & Wang (2023) proposed a family of methods that automatically leverage task similarities to improve multitask learning. These approaches are orthogonal to our method and can be potentially combined with the data and task selection enabled by our method. There is also a body of work on task relatedness in transfer learning (Zamir et al., 2018; Achille et al., 2021; Dwivedi & Roig, 2019; Zhuang et al., 2021; Achille et al., 2019). However, Standley et al. (2020) demonstrated that task similarity metrics in transfer learning do not generalize well to the multitask learning domain. Finally, various works have tackled negative transfer in MTL from different perspectives. Yang et al. (2019) propose learning latent task representations and utilizing a block-diagonal latent task assignment matrix to explicitly structure inter-task relationships, thereby mitigating inter-group negative transfer. Meng et al. (2021) reformulate the MTL problem as a multi-teacher knowledge distillation framework, introducing a two-stage optimization process that alleviates negative transfer. Zhou et al. (2023a) address negative transfer by introducing the Feature Decomposition Network (FDN), which separates features into task-specific and task-shared components to reduce feature redundancy. These works address related topics and are complementary to ours, included here for completeness in reviewing the literature.

## 3 PRELIMINARY: INFLUENCE FUNCTION AS AN APPROXIMATION TO LOO

As a widely used data attribution metric, the leave-one-out (LOO) effect measures the contribution of a training data point by the change of model performance after removing this data point and retraining the model (Koh & Liang, 2017; Schioppa et al., 2022; Grosse et al., 2023). However, repeatedly retraining the model can be computationally extensive. To address this issue, in the single-task learning (STL) setting, Koh & Liang (2017) proposed the use of influence functions, which approximate the LOO effect by leveraging small perturbations to the weight of the loss at each data point.

Specifically, for a given data point $z \in \mathcal{Z}$ and parameter vector $\theta \in \Theta$, consider a loss function $\ell(\theta; z)$. Given a training dataset $\{z_i\}_{i=1}^n$, we minimize the empirical risk, i.e., $\hat{\theta} = \arg\min_{\theta \in \Theta} \sum_{i=1}^n \ell(\theta; z_i)/n$, and evaluate the performance of $\hat{\theta}$ using certain evaluation metrics. A common metric is the average loss on the validation data $D^v$, i.e., $V(\hat{\theta}; D^v) = \sum_{z \in D^v} \ell(\hat{\theta}; z)/|D^v|$. The LOO effect of the $i$-th data point is defined as the difference in the evaluation metric when using the parameters learned from all data points versus the parameters learned by excluding the data point $z_i$. Formally, we introduce a weight vector $\boldsymbol{\sigma} = (\sigma_1, \cdots, \sigma_n)$ into the objective function, then the minimizer can be written by

$$\hat{\theta}(\boldsymbol{\sigma}) = \arg\min_{\theta \in \Theta} \mathcal{L}(\theta, \boldsymbol{\sigma}), \text{ where } \mathcal{L}(\theta, \boldsymbol{\sigma}) := \frac{1}{n} \sum_{i=1}^n \sigma_i \ell(\theta; z_i).$$

The LOO effect of the $i$-th data point is given by $V(\hat{\theta}(\mathbf{1}); D^v) - V(\hat{\theta}(\mathbf{1}^{(-i)}); D^v)$, where $\mathbf{1}$ is an all-ones vector with length $n$ and $\mathbf{1}^{(-i)}$ is a vector of all ones except for the $i$-th element being 0. The LOO effect requires retraining the model multiple times — once for each data point being left out - to obtain $\hat{\theta}(\mathbf{1}^{(-i)})$. To reduce computational cost, Koh & Liang (2017) proposed to approximate the

LOO effect by using the partial derivative $\nabla_\theta V(\hat{\theta}(\boldsymbol{\sigma}); D^{\mathrm{v}})^\top \cdot \frac{\partial \hat{\theta}(\boldsymbol{\sigma})}{\partial \sigma_i}\Big|_{\boldsymbol{\sigma}=\mathbf{1}}$. Under certain regularity conditions, the effect of perturbing the weight for data point $z_i$ on the learned parameters is given by

$$\frac{\partial \hat{\theta}(\boldsymbol{\sigma})}{\partial \sigma_i}\Big|_{\boldsymbol{\sigma}=\mathbf{1}} = -H(\hat{\theta}(\mathbf{1}), \mathbf{1})^{-1} \cdot \nabla_\theta \ell(\hat{\theta}(\mathbf{1}); z_i), \tag{1}$$

where $H(\theta, \boldsymbol{\sigma}) = \sum_{i=1}^n \sigma_i \frac{\partial^2 \ell(\theta; z_i)}{\partial\theta\partial\theta^\top}/n$ is the Hessian matrix. (See Appendix A of Koh & Liang (2017) for the derivation.) This approximation is referred to as influence function (IF)-based data attribution. Compared to the LOO effect, IF-based data attribution only requires the evaluation of the inverse Hessian matrix and the gradient at the model parameters trained on the full dataset.

## 4 INFLUENCE FUNCTION FOR MULTITASK DATA ATTRIBUTION

While IF-based data attribution has been shown as a scalable and effective tool for many applications, it has been primarily developed for STL settings, where a single model is trained on a homogeneous task. However, in many real-world applications, multiple related tasks are learned jointly, with shared parameters across tasks and different evaluation metrics of interest. In this section, we extend IF-based data attribution to the multitask learning (MTL) setting.

### 4.1 PROBLEM SETUP FOR MULTITASK DATA ATTRIBUTION

**Multitask Learning.** MTL aims to solve multiple tasks simultaneously. In many real-world scenarios, tasks are often related and share common underlying structures. MTL leverages shared structures by jointly training tasks to enhance generalization and improve prediction accuracy, especially when tasks are related or when data for individual tasks is limited. A common approach in MTL to facilitate information sharing across tasks is through either soft or hard parameter sharing (Ruder, 2017). In soft parameter sharing, regularization is applied to the task-specific parameters to encourage them to be similar across tasks (Xue et al., 2007; Duong et al., 2015). In contrast, hard parameter sharing learns a common feature representation through shared parameters, while task-specific parameters are used to make predictions tailored to each task (Caruana, 1997). Recently, Duan & Wang (2023) proposed an augmented optimization framework for MTL that accommodates both hard parameter sharing and various types of soft parameter sharing.

We consider a general MTL objective that incorporates these common parameter-sharing schemes. Specifically, consider $K$ tasks and for each task $k = 1, \ldots, K$, we observe $n_k$ independent samples, denoted by $\{z_{ki}\}_{i=1}^{n_k}$. Let $\ell_k(\cdot; \cdot)$ be the loss function for task $k$. The MTL objective is given by

$$\mathcal{L}(\boldsymbol{w}) = \sum_{k=1}^K \left[ \frac{1}{n_k} \sum_{i=1}^{n_k} \ell_k(\theta_k, \gamma; z_{ki}) + \Omega_k(\theta_k, \gamma) \right], \tag{2}$$

where $\boldsymbol{\theta} = \{\theta_k \in \mathbb{R}^{d_k}\}_{k=1}^K$ are task-specific parameters, $\gamma \in \mathbb{R}^p$ are shared parameters, $\boldsymbol{w} = \{\boldsymbol{\theta}, \gamma\}$ denotes all parameters, and $\Omega_k(\theta_k, \gamma)$ represents the task-level regularization. The parameters are estimated by minimizing (2), i.e., $\hat{\boldsymbol{w}} = \arg\min_{\boldsymbol{w}} \mathcal{L}(\boldsymbol{w})$.

Below, we present two special cases of supervised learning within this general framework: one illustrating soft parameter sharing and the other demonstrating hard parameter sharing. Let $z_{ki} = (x_{ki}, y_{ki})$ for $1 \le k \le K$ and $1 \le i \le n_k$, where $x_{ki}$ represents the features and $y_{ki}$ represents the outcomes for the $i$-th data point in task $k$.

**Example 1 (Multitask Linear Regression with Ridge Penalty).** *Regularization has been integrated in MTL to encourange similarity among task-specific parameters; see (Evgeniou & Pontil, 2004; Duan & Wang, 2023) for examples. Consider the regression setting where $y_{ki} = x_{ki}^\top \theta_k^* + \epsilon_{ki}$, with $\epsilon_{ki}$ being independent noise and $x_{ki} \in \mathbb{R}^d$ for $1 \le i \le n_k$ and $1 \le k \le K$. Additionally, we have the prior knowledge that $\{\theta_k^*\}_{k=1}^K$ are close to each other. Instead of fitting a separate ordinary least squares estimator for each $\theta_k$, a ridge penalty is introduced to shrink the task-specific parameters $\theta_1, \ldots, \theta_K \in \mathbb{R}^d$ toward a common vector $\gamma \in \mathbb{R}^d$, while $\gamma$ is simultaneously learned by leveraging data from all tasks.*

*The objective function for multitask linear regression with a ridge penalty is given by*

$$\mathcal{L}(\boldsymbol{w}) = \sum_{k=1}^K \left[ \frac{1}{n_k} \sum_{i=1}^{n_k} (y_{ki} - x_{ki}^\top \theta_k)^2 + \lambda_k \|\theta_k - \gamma\|_2^2 \right],$$

where $\lambda_k$ controls the strength of regularization. This can be viewed as a special case of (2) by setting $\ell_k$ as the squared error (depending only on the task-specific parameters) and defining the regularization term $\Omega_k(\theta_k, \gamma) = \lambda_k \|\theta_k - \gamma\|_2^2$.

**Example 2 (Shared-Bottom Neural Network Model).** *The shared-bottom neural network architecture, first proposed by Caruana (1997), has been widely applied to MTL across various domains (Zhou et al., 2023b; Liu et al., 2021b; Ma et al., 2018). The shared-bottom model can be represented as $f_k(x) = g(\theta_k; f(\gamma; x))$, where $f(\gamma; \cdot)$ represents the shared layers that process the input data and produce an intermediate representation, and $\gamma$ denotes the parameters shared across tasks. The function $g(\theta_k; \cdot)$ corresponds to task-specific layers, which take the intermediate representation and produce task-specific predictions, with $\theta_k$ representing task-specific parameters.*

*The loss function for this model can be written as:*

$$\mathcal{L}(\boldsymbol{w}) = \sum_{k=1}^{K} \left[ \frac{1}{n_k} \sum_{i=1}^{n_k} \ell_k(y_{ki}, g(\theta_k; f(\gamma; x_{ki}))) + \Omega_k(\theta_k, \gamma) \right],$$

*where $\ell_k(\cdot, \cdot)$ represents the task-specific loss function, and $\Omega_k(\theta_k, \gamma)$ denotes the regularization term. A simple choice is $\Omega_k(\theta_k, \gamma) = \lambda_k(\|\theta_k\|_2^2 + c\|\gamma\|_2^2)$, where $\lambda_k$ and $c$ are positive constants.*

**Multitask Data Attribution.** In this work, we aim to estimate the contribution of a data point (or a task) to the learning performance on a specific target task $k \in \{1, \ldots, K\}$. The performance of any model with parameters $(\theta_k, \gamma)$ on task $k$ can be measured by the average loss over a validation dataset $D_k^{\mathrm{v}}$, i.e, $V_k(\theta_k, \gamma; D_k^{\mathrm{v}}) = \sum_{z \in D_k^{\mathrm{v}}} \ell_k(\theta_k, \gamma; z)/|D_k^{\mathrm{v}}|$. Then the *data-level influence* of the $i$-th data point from task $l$ on the target task $k$ can be quantified by the following LOO effect:

$$\Delta_k^{li} := V_k(\hat{\theta}_k, \hat{\gamma}; D_k^{\mathrm{v}}) - V_k(\hat{\theta}_k^{(-li)}, \hat{\gamma}^{(-li)}; D_k^{\mathrm{v}}), \tag{3}$$

where $\hat{\theta}_k$ and $\hat{\gamma}$ are from the minimizer of (2) with the full training data, while $\hat{\theta}_k^{(-li)}$ and $\hat{\gamma}^{(-li)}$ are obtained by excluding the data point $z_{li}$ from task $l$. This data-level attribution metric allows for a fine-grained understanding of the impact each data point from one task has on another task.

Similarly, the *task-level influence* of task $l$ on the target task $k$ is quantified by the leave-one-task-out (LOTO) effect:

$$\Delta_k^{l} := V_k(\hat{\theta}_k, \hat{\gamma}; D_k^{\mathrm{v}}) - V_k(\hat{\theta}_k^{(-l)}, \hat{\gamma}^{(-l)}; D_k^{\mathrm{v}}), \tag{4}$$

where $\hat{\theta}_k^{(-l)}$ and $\hat{\gamma}^{(-l)}$ are obtained by excluding all the data points from task $l$. The LOTO effect provides a natural and interpretable measure of task relatedness.

### 4.2 THE PROPOSED METHOD: MULTITASK INFLUENCE FUNCTION

The computational burden of evaluating LOO and LOTO effects becomes even more pronounced in MTL setting compared to STL setting, particularly when the number of tasks is large. To address this challenge, we extend the IF-based approximation to LOO and LOTO effects in MTL. This approach builds on the similar idea of using infinitesimal perturbations on the weights of data points to approximate the removal of individual data points. Specifically, we consider the following data-level $\boldsymbol{\sigma}$-weighted version of the general objective function in (2):

$$\mathcal{L}(\boldsymbol{w}, \boldsymbol{\sigma}) = \sum_{k=1}^{K} \left[ \frac{1}{n_k} \sum_{i=1}^{n_k} \sigma_{ki} \ell_{ki}(\theta_k, \gamma) + \Omega_k(\theta_k, \gamma) \right], \tag{5}$$

where $\ell_{ki}(\cdot)$ is shorthand for $\ell_k(\cdot; z_{ki})$. For each weight vector $\boldsymbol{\sigma}$, we solve $\boldsymbol{w}(\boldsymbol{\sigma}) = \arg\min_{\boldsymbol{w}} \mathcal{L}(\boldsymbol{w}, \boldsymbol{\sigma})$. We propose to use the partial derivative with respect to $\sigma_{li}$, i.e.,

$$\left. \frac{\partial V_k(\hat{\theta}_k(\boldsymbol{\sigma}), \hat{\gamma}(\boldsymbol{\sigma}); D_k^{\mathrm{v}})}{\partial \sigma_{li}} \right|_{\boldsymbol{\sigma}=\mathbf{1}} = \nabla_\theta V_k(\hat{\theta}_k, \hat{\gamma}; D_k^{\mathrm{v}}) \cdot \left. \frac{\partial \hat{\theta}_k(\boldsymbol{\sigma})}{\partial \sigma_{li}} \right|_{\boldsymbol{\sigma}=\mathbf{1}} + \nabla_\gamma V_k(\hat{\theta}_k, \hat{\gamma}; D_k^{\mathrm{v}}) \cdot \left. \frac{\partial \hat{\gamma}(\boldsymbol{\sigma})}{\partial \sigma_{li}} \right|_{\boldsymbol{\sigma}=\mathbf{1}},$$
$$\tag{6}$$

to approximate the LOO effect defined in (3). To apply the chain rule in (6), we need to compute the influence scores of the data point $z_{li}$ on the task-specific parameters $\hat{\theta}_k$ and shared parameters $\hat{\gamma}$.

To achieve this, we present the following proposition that provides the explicit analytical form for the influence of a data point on task-specific parameters for the same task (within-task influence), task-specific parameters for another task (between-task influence), and shared parameters (shared influence). Before introducing the results, we first define some notation. Let $H_{kl}$ denote the $(k, l)$-th

block components of the Hessian matrix of the MTL objective function $\mathcal{L}(\boldsymbol{w}, \boldsymbol{\sigma})$, as defined in (5), with respect to $\boldsymbol{w}$. This Hessian matrix has the following *block structure* in MTL:

$$H(\boldsymbol{w}, \boldsymbol{\sigma}) = \begin{pmatrix} H_{1,1} & \cdots & 0 & H_{1,K+1} \\ \vdots & \ddots & \vdots & \vdots \\ 0 & \cdots & H_{K,K} & H_{K,K+1} \\ H_{K+1,1} & \cdots & H_{K+1,K} & H_{K+1,K+1} \end{pmatrix}. \tag{7}$$

The details of each block are described in Lemma A.1. We leverage the unique block structure of this Hessian in MTL to derive its analytical inverse, offering insights into how data from other tasks influence the target task through shared parameters.

**Proposition 1** (Data-Level Within-task Influence, Between-task Influence, and Shared Influence).
*Assuming the objective function $\mathcal{L}(\boldsymbol{w}, \boldsymbol{\sigma})$ in (5) is twice-differentiable and strictly convex in $\boldsymbol{w}$. For any two tasks $k \neq l$ and $1 \leq k, l \leq K$, the following results hold:*

*(Shared influence) For $1 \leq i \leq n_k$, the influence of the $i$-th data point from task $k$ on the shared parameters, $\hat{\gamma}$, is given by*

$$\frac{\partial \hat{\gamma}}{\partial \sigma_{ki}} = N^{-1} \cdot H_{K+1,k} H_{kk}^{-1} \frac{\partial \ell_{ki}}{\partial \theta_k} - N^{-1} \frac{\partial \ell_{ki}}{\partial \gamma}, \tag{8}$$

*where the matrix $N := H_{K+1,K+1} - \sum_{k=1}^{K} H_{K+1,k} H_{kk}^{-1} H_{k,K+1} \in \mathbb{R}^{p \times p}$ is invertible;*

*(Within-task influence) For $1 \leq i \leq n_k$, the influence of the $i$-th data point from task $k$ on the task-specific parameters for the same task $k$, $\hat{\theta}_k$, is given by*

$$\frac{\partial \hat{\theta}_k}{\partial \sigma_{ki}} = -H_{kk}^{-1} \frac{\partial \ell_{ki}}{\partial \theta_k} - H_{kk}^{-1} H_{k,K+1} \cdot \frac{\partial \hat{\gamma}}{\partial \sigma_{ki}}; \tag{9}$$

*(Between-task influence) For $1 \leq i \leq n_l$, the influence of the $i$-th data point from task $l$ on the task-specific parameters for another task $k$, $\hat{\theta}_k$, is given by*

$$\frac{\partial \hat{\theta}_k}{\partial \sigma_{li}} = -H_{kk}^{-1} H_{k,K+1} \cdot \frac{\partial \hat{\gamma}}{\partial \sigma_{li}}. \tag{10}$$

**Interpretation of Data-Level Influences.** In MTL, data points have more composite influences on task-specific parameters compared to STL due to interactions with other tasks and shared parameters. In STL, each data point only affects its own task's parameters through the gradient and Hessian of the task-specific objective, which is solely the first term in (9). However, in MTL, shared parameters introduce a feedback mechanism that allows data from one task to influence the parameters of other tasks. As shown in (8), the influence of $i$-th data point from task $k$ on the shared parameters stem from two sources: the first term reflects the change on the task-specific parameter $\hat{\theta}_k$, which then indirectly affects the shared parameters $\hat{\gamma}$, while the second term accounts for the direct impact on $\hat{\gamma}$. Consequently, within-task influence in (9) includes an additional influence propagated through the shared parameters, and between-task influence in (10) arises as data from one task indirectly impacts the parameters of another task via the shared parameters. In particular, in STL, between-task influence does not occur because tasks are independent and do not interact.

**Computational Complexity of Exact Hessian Inverse Calculation.** Evaluating influence scores requires inverting the Hessian matrix, which becomes computationally expensive as the number of parameters per task or the number of tasks increases. The exact calculation of $H^{-1}(\boldsymbol{w}, \boldsymbol{\sigma})$ has a complexity of $\Omega\left(\left(\sum_{k=1}^{K} d_k + p\right)^w\right)$, where $w$ is the matrix multiplication constant. The current state-of-the-art value is $w \approx 2.37$ (Le Gall, 2014). However, using the block structure of the Hessian matrix in Proposition 1, the complexity can be largely reduced to $\Omega\left(\sum_{k=1}^{K} d_k^w + p^w\right)$.

**Challenges in Approximating Hessian Inverse in MTL.** Recent methods, such as EK-FAC (George et al., 2018; Grosse et al., 2023) and LiSSA (Agarwal et al., 2017; Koh & Liang, 2017), have been developed to approximate the Hessian inverse and improve computational efficiency. We discuss the challenges these methods may face when applied to MTL, highlighting the need for further exploration and adaptation to better align with the block-structure of Hessian in MTL.

**EK-FAC** approximates the Hessian inverse using a blockwise diagonal matrix by treating different layers of a neural network as independent. For many popular MTL models (e.g., the shared-bottom

neural network model in Example 2), EK-FAC approximates the Hessian matrix in Equation (7) as $\text{diag}\{H_{1,1}, \ldots, H_{K+1,K+1}\}$, where the off-diagonal blocks $H_{k,K+1}$ and $H_{K+1,k}$ are treated as $\mathbf{0}$ for all $k = 1, \ldots, K$. However, as shown in Proposition 1, this approximation can result in the loss of significant contributions to within-task influence, between-task influence, and shared influence in MTL due to omitting the terms involving $H_{k,K+1}$ and $H_{K+1,k}$. For instance, under this approximation, $\frac{\partial \hat{\theta}_k}{\partial \sigma_{li}}$ would be approximated as $0$, which can severely degrade the accuracy of between-task influence score calculations. This limitation is particularly impactful in soft parameter sharing models, where the influence from one task to another will be completely lost.

**LiSSA** approximates the *inverse-Hessian-vector-product* using an iterative algorithm that supports mini-batch gradients. While LiSSA has potential applicability in the MTL setting, its convergence heavily depends on the condition number of the empirical Hessian evaluated on each mini-batch (Agarwal et al., 2017). The condition number of a matrix, defined as the ratio of its largest eigenvalue to its smallest eigenvalue, quantifies the matrix's ill-posedness. In the MTL setting, each data point is associated with a specific task $k$, influencing the shared parameters and only the task-specific parameters for task $k$. Consequently, the empirical Hessian for a single data point contains non-zero entries only in the $H_{k,k}$, $H_{k,K+1}$, $H_{K+1,k}$, and $H_{K+1,K+1}$ blocks of Equation (7). This sparsity can render the mini-batch empirical Hessian ill-posed, leading to a large condition number and instability in LiSSA's iterative approximation.

In Appendix C.4, we empirically demonstrate that as the number of tasks $K$ increases, LiSSA requires larger batch sizes in the stochastic estimation to mitigate the issues caused by a high condition number. This adjustment significantly increases computational costs for large-scale MTL problems.

### 4.3 TASK-LEVEL INFLUENCES

The LOTO effect, introduced in (4) is a natural measure for task relatedness. To provide a computationally efficient approximation of the LOTO effect, we similarly apply infinitesimal perturbations on the data weights. Specifically, we consider the following task-level $\boldsymbol{\sigma}$-weighted objective, where we assign the same weight to data from the same task:

$$\mathcal{L}(\boldsymbol{w}, \boldsymbol{\sigma}) = \sum_{k=1}^{K} \sigma_k \left[ \frac{1}{n_k} \sum_{i=1}^{n_k} \ell_{ki}(\theta_k, \gamma) + \Omega_k(\theta_k, \gamma) \right]. \tag{11}$$

Note that, the regularization terms $\Omega_k(\theta_k, \gamma)$ in (11) are also weighted by $\sigma_k$, unlike the data-level $\boldsymbol{\sigma}$-weighted objective (5), where the weights are only applied to the individual losses $l_{ki}(\theta_k, \gamma)$. This difference is due to the nature of MTL - excluding a task results in the removal of its task-specific parameters along with the regularization term.

The IF-based approximation for the LOTO effect $\Delta_k^l$ is given by

$$\frac{\partial V_k(\hat{\theta}_k(\boldsymbol{\sigma}), \hat{\gamma}(\boldsymbol{\sigma}); D_k^{\mathrm{v}})}{\partial \sigma_l} \bigg|_{\boldsymbol{\sigma}=\mathbf{1}} = \nabla_\theta V_k(\hat{\theta}_k, \hat{\gamma}; D_k^{\mathrm{v}}) \cdot \frac{\partial \hat{\theta}_k(\boldsymbol{\sigma})}{\partial \sigma_l} \bigg|_{\boldsymbol{\sigma}=\mathbf{1}} + \nabla_\gamma V_k(\hat{\theta}_k, \hat{\gamma}; D_k^{\mathrm{v}}) \cdot \frac{\partial \hat{\gamma}(\boldsymbol{\sigma})}{\partial \sigma_l} \bigg|_{\boldsymbol{\sigma}=\mathbf{1}}.$$

In Proposition 2, we provide the analytical form for the influence of data from one task on the parameters of another task and the shared parameters. The Hessian matrix of $\mathcal{L}(\boldsymbol{w}, \boldsymbol{\sigma})$ with respect to $\boldsymbol{w}$ shares the same block structure as shown in (7). Let $H_{kl}$ denote the $(k, l)$-th block of the Hessian matrix, with the details provided in Lemma A.2. Let $N$ be defined as in Proposition 1.

**Proposition 2** (Task-Level Between-task Influence)**.** *Under the assumptions of Proposition 1, for any two tasks $k \neq l$ where $1 \leq k, l \leq K$, the influence of data from task $l$ on the task-specific parameters of task $k$, $\hat{\theta}_k$, is given by*

$$\frac{\partial \hat{\theta}_k}{\partial \sigma_l} = -H_{kk}^{-1} H_{k,K+1} \cdot \frac{\partial \hat{\gamma}}{\partial \sigma_l}, \tag{12}$$

*where $\frac{\partial \hat{\gamma}}{\partial \sigma_l}$ is the influence of data from task $l$ on the shared parameters, $\hat{\gamma}$, and is given by*

$$\frac{\partial \hat{\gamma}}{\partial \sigma_l} = N^{-1} H_{K+1,l} H_{ll}^{-1} \left[ \sum_{i=1}^{n_l} \frac{\partial \ell_{li}}{\partial \theta_l} + \frac{\partial \Omega_l}{\partial \theta_l} \right] - N^{-1} \left[ \sum_{i=1}^{n_l} \frac{\partial \ell_{li}}{\partial \gamma} + \frac{\partial \Omega_l}{\partial \gamma} \right]. \tag{13}$$

As shown in Proposition 2, task-level influences $\frac{\partial \hat{\theta}_k}{\partial \sigma_l}$ and $\frac{\partial \hat{\gamma}}{\partial \sigma_l}$ are sums of data-level influence scores for all points in task $l$, with additional terms arising from $\sigma$-weighted regularization.

# 5 EXPERIMENTS

Our experimental results are organized into two parts: (1) an evaluation of MTIF's ability to approximate the gold-standard LOO and LOTO influences, and (2) a demonstration of its practical application for data selection to improve model performance.

We test MTIF on two types of models: linear models, illustrated in Example 1 and discussed in Section 5.1, and neural network models, described in Example 2 and discussed in Section 5.2. The results highlight the following: First, our data-level influence score provides a strong approximation of the leave-one-out (LOO) effect in (3), as evidenced by the high degree of linearity between the two measures. Second, our task-level influence scores closely approximate the gold-standard LOTO influences in (4), as demonstrated by a high Spearman correlation between the ranks of task contributions derived from our method and those obtained via true LOTO influences. Additionally, in Appendix C.2, we compare MTIF with gradient-based baseline methods (Azorin et al., 2023; Fifty et al., 2021)for measuring task relatedness between training tasks and the target task. Our results show that the Spearman correlation between the task-relatedness ranks estimated by baseline methods and the gold-standard LOTO ranks is consistently lower than that achieved by MTIF.

For practical applications, in Section 5.3, we show that data selection based on MTIF influence scores enhances model performance across various dynamic weighting algorithms (Liu et al., 2019; Chen et al., 2018; Liu et al., 2021a; Kendall et al., 2018; Lin et al., 2022) for MTL. Furthermore, in Appendix D, we present case studies demonstrating that the negative samples filtered out by MTIF provide interpretable insights into why they are detrimental to the model's performance.

## 5.1 LINEAR MODEL

We conduct experiments in the linear model setting, as described in Example 1, on two datasets.
**Synthetic Dataset.** This dataset consists of 10 tasks, each with 200 samples $(x_{ji}, y_{ji})$ split equally into training and testing sets. Input vectors $x_{ji}$ are drawn from $\mathcal{N}(0, I_d)$ with $d = 50$, and responses are generated using $y_{ji} = x_{ji}^\top \theta_j^\star + \epsilon_{ji}$, where $\epsilon_{ji} \sim \mathcal{N}(0, 1)$. More details are provided in Appendix C.1.1.
**Real-World Dataset.** The second dataset is the HAR dataset (Anguita et al., 2013), also referenced in Duan & Wang (2023). This dataset was collected from 30 volunteers performing daily activities while carrying a smartphone equipped with inertial sensors on their waist. To adapt this dataset for multitask learning, we treat the data from each volunteer as a separate task, with the objective of distinguishing sitting from all other activities. Additional details can be found in Appendix C.1.2.
We apply linear regression on the synthetic dataset and logistic regression on the HAR dataset. We present experimental results for data-level and task-level influence scores derived from our method, evaluated on both datasets.

**Data-level Influence.** In this experiment, we compare our data-level influence scores (6) with the gold-standard LOO scores (3). The results, shown in Figure 1, reveal a strong linear correlation between the MTIF influence scores and the gold-standard LOO scores across all scenarios. This demonstrates that MTIF effectively approximates the LOO effect for both within-task and between-task influences on the synthetic and HAR datasets.

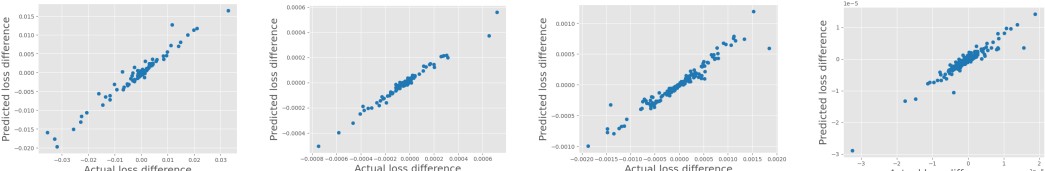

Figure 1: LOO experiments on linear models. The x-axis is the actual loss difference obtained by LOO retraining, and the y-axis is the predicted loss difference calculated by MTIF. The first two figures from the left show within-task and between-task results (in order) results on the synthetic dataset, while the other two figures present within-task and between-task results (in order) on the HAR dataset. Following Koh & Liang (2017), the plots shown here reflect influences on a randomly picked test data point, while the trend holds more broadly on other test data points. The scatter points correspond to training data points in the first task of each dataset.

**Task-level Influence.** In this experiment, we compare our task-level influence scores with the gold-standard LOTO scores (4) on both the synthetic and HAR datasets. We randomly split 20% of

the data from each task as the validation set. Specifically, for a given target task, we use MTIF to calculate the influence score of each task on the model's performance on the target task's validation set. We also compute the LOTO scores for each task by retraining the model. The Spearman correlation coefficient between the MTIF influence scores and the LOTO scores is then reported.

Table 1 shows the results on the synthetic dataset for each task selected as the target task. Due to space constraints, the results for the HAR dataset are provided in Appendix C.1.2. On both datasets, the proposed MTIF achieves high correlation coefficients with the LOTO scores, indicating that MTIF aligns well with LOTO. Additionally, we compare MTIF with two baselines, Cosine Similarity (Azorin et al., 2023) and TAG (Fifty et al., 2021), in Appendix C.2. These baselines are widely used for measuring task relatedness in the MTL literature. The proposed MTIF outperforms these baselines in terms of correlation with the ground-truth LOTO scores.

| Task 1 | Task 2 | Task 3 | Task 4 | Task 5 |
|---|---|---|---|---|
| $0.84 \pm 0.05$ | $0.72 \pm 0.05$ | $0.74 \pm 0.11$ | $0.81 \pm 0.05$ | $0.71 \pm 0.09$ |
| Task 6 | Task 7 | Task 8 | Task 9 | Task 10 |
| $0.74 \pm 0.04$ | $0.74 \pm 0.07$ | $0.84 \pm 0.03$ | $0.74 \pm 0.03$ | $0.65 \pm 0.07$ |

Table 1: Average Spearman correlation coefficients across 5 random seeds on the synthetic dataset. Error bars represent the standard error of the mean.

## 5.2 Neural Networks

We further evaluate the effectiveness of the proposed MTIF in neural network settings with the Shared-Bottom neural network (as described in Example 2) on the CelebA dataset (Liu et al., 2015). In our experiments, the Shared-Bottom model consists of a two-layer convolutional neural network as the shared encoder and a single-layer feed-forward neural network as the task-specific decoder for each task. Details of the training procedure are provided in Appendix C.1.3.

**CelebA Dataset.** CelebA is a large-scale face image dataset annotated with 40 attributes and widely used in the multitask learning (MTL) literature (Fifty et al., 2021). Following the setup in Fifty et al. (2021), we select 9 attributes as tasks for our experiments, modeling each task as a binary classification problem.

**Task-level Influence.** In this experiment, we compare the task-level influence estimated by MTIF with the gold-standard LOTO scores in the neural network setting, following a similar setup as the linear model experiments in Section 5.1. Table 2 reports the average Spearman correlation coefficients across 5 random seeds, with each task selected as the target task. Compared to the linear model setting in Table 1, the correlation coefficients are lower. This result is expected, as data attribution for non-convex models is inherently more challenging, and the evaluation is noisier due to the stochasticity in model retraining (Koh & Liang, 2017).

Despite these challenges, the influence scores estimated by MTIF still exhibit non-trivial correlations with the LOTO scores in most cases, with the highest correlation coefficient reaching 0.43. This demonstrates that MTIF effectively captures meaningful signals even in the neural network setting. Furthermore, MTIF outperforms baseline task-relatedness measures in this setup, as detailed in Appendix C.2.

| Task 1 | Task 2 | Task 3 | Task 4 | Task 5 | Task 6 | Task 7 | Task 8 | Task 9 |
|---|---|---|---|---|---|---|---|---|
| $0.21 \pm 0.04$ | $0.35 \pm 0.19$ | $0.23 \pm 0.10$ | $0.43 \pm 0.12$ | $0.14 \pm 0.17$ | $0.36 \pm 0.10$ | $0.29 \pm 0.05$ | $0.10 \pm 0.10$ | $0.15 \pm 0.15$ |

Table 2: Average Spearman correlation coefficients across 5 random seeds on the CelebA dataset with the Shared-Bottom Neural Network model. Error bars represent the standard error of the mean.

## 5.3 Application: Data Selection

We further illustrate the practical utility of the proposed MTIF through a downstream application in data selection. While most existing MTL research focuses on task-level relatedness, the data-level influence scores estimated by MTIF offer a unique opportunity to improve MTL performance by identifying and removing training data points that negatively impact the model. In this section, we present experiment results on the CelebA dataset using the Shared-Bottom Neural Network architecture. Additional results on simpler linear models as well as results on other MTL model architectures beyond shared-bottom neural networks, are included in Appendix C.3.

**Experimental Setup.** To evaluate the compatibility of MTIF's data selection with various MTL algorithms and to compare their performances, we conduct experiments using different MTL training algorithms. Specifically, in addition to the naive Equal Weighting (EW) training algorithm, we evaluate several re-weighting-based methods: GradNorm (GN) (Chen et al., 2018), Dynamic Weight

|         | Task 1 | Task 2 | Task 3 | Task 4 | Task 5 | Task 6 | Task 7 | Task 8 | Task 9 | Average |
|---------|--------|--------|--------|--------|--------|--------|--------|--------|--------|---------|
| DWA     | 0.864† | 0.804† | 0.903† | 0.786† | 0.931† | 0.945* | 0.854† | 0.920* | 0.955† | 0.885*  |
| DWA+DS  | 0.873† | 0.815† | 0.904† | 0.795† | 0.937† | 0.950* | 0.866† | 0.929* | 0.956† | 0.892*  |
| GN      | 0.864† | 0.808† | 0.900† | 0.781† | 0.928† | 0.946† | 0.856† | 0.922† | 0.954† | 0.884*  |
| GN+DS   | 0.873† | 0.817† | 0.898† | 0.791† | 0.934† | 0.951* | 0.870† | 0.931* | 0.957† | 0.891*  |
| IMTL    | 0.858† | 0.800† | 0.896† | 0.775† | 0.930† | 0.947* | 0.869† | 0.917† | 0.957† | 0.883*  |
| IMTL+DS | 0.871* | 0.806† | 0.897† | 0.788† | 0.934† | 0.952* | 0.873† | 0.925† | 0.961† | 0.890*  |
| UW      | 0.857† | 0.808† | 0.897† | 0.781† | 0.925† | 0.945* | 0.859† | 0.920* | 0.956† | 0.883*  |
| UW+DS   | 0.867† | 0.816† | 0.900† | 0.792† | 0.931† | 0.953* | 0.866† | 0.929† | 0.958† | 0.890*  |
| RLW     | 0.870† | 0.821† | 0.901† | 0.789† | 0.934† | 0.951† | 0.856† | 0.927* | 0.955† | 0.889*  |
| RLW+DS  | 0.881* | 0.820† | 0.907† | 0.798† | 0.936† | 0.954* | 0.868* | 0.932* | 0.958† | 0.895*  |
| EW      | 0.861† | 0.813† | 0.901† | 0.784† | 0.927† | 0.946† | 0.856† | 0.918* | 0.957† | 0.885*  |
| EW+DS   | 0.869† | 0.818† | 0.902† | 0.792† | 0.934† | 0.952* | 0.868† | 0.929† | 0.958† | 0.892*  |

Table 3: Results of model performance using different dynamic weighting methods (DWA (Liu et al., 2019), GN (Chen et al., 2018), IMTL (Liu et al., 2021a), UW (Kendall et al., 2018), and RLW (Lin et al., 2022)), both with and without data selection (DS). The DS method removes the top 5% of the most negative data points based on the data-level influence scores estimated by MTIF. EW refers to Equal Weighting, which is the vanilla Shared-Bottom model without any re-weighting. The reported values are averaged over 5 random seeds, with † indicating standard error of the mean $< 0.01$ and * indicating standard error of the mean $< 0.002$. The last column shows the average performance across all tasks.

Average (DWA) (Liu et al., 2019), Impartial MultiTask Learning (IMTL) (Liu et al., 2021a), Random Loss Weighting (RLW) (Lin et al., 2022), and Uncertainty Weighting (UW) (Kendall et al., 2018). These methods dynamically measure and adjust task relatedness during training.

We assess model performance under two configurations: (1) using the MTL algorithms as standalone methods, and (2) combining these algorithms with data selection (DS) enabled by MTIF. For data selection, the model is initially trained, after which the top 5% of the most detrimental training data points, as estimated by MTIF on the validation dataset, are removed. The models are then retrained on the remaining training data. All re-weighting methods and their combinations with data selection are implemented using libMTL (Lin & Zhang, 2023).

**Results.** Table 3 summarizes our key findings. First, the standalone re-weighting-based methods provide minimal to no improvement over the baseline EW, with only RLW (0.889) slightly outperforming EW (0.885) in terms of average performance. Although this result may seem counterintuitive, it aligns with findings from the benchmark study by libMTL (Lin & Zhang, 2023). Second, introducing data selection significantly enhances the performance of the baseline, with EW+DS achieving an average performance of 0.892, outperforming all standalone re-weighting-based methods. Third, combining data selection with re-weighting-based methods consistently results in substantial performance gains compared to their standalone counterparts.

These findings demonstrate that leveraging fine-grained data-level influence through MTIF-driven data selection provides more substantial improvements in MTL performance than approaches that focus solely on task-level relatedness.

## 6 CONCLUSION

In this work, we proposed the *MultiTask Influence Function* (**MTIF**), a novel data attribution method for multitask learning (MTL). MTIF efficiently estimates the influence of individual data points on task performance across multiple tasks, without the need for retraining. By leveraging the structure of MTL models, MTIF enables scalable and interpretable data-level and task-level influence analysis.

Through experiments on both linear and neural network models, we demonstrated that MTIF effectively approximates leave-one-out and leave-one-task-out effects. Moreover, MTIF facilitates fine-grained data selection, leading to consistent improvements in model performance and helping mitigate negative transfer. Our approach offers a practical tool for measuring and interpreting task relatedness as well as improving MTL performance.

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

## A    LEMMA

The first two lemmas describe the structure of the Hessian matrices for data-level and task-level inference.

**Lemma A.1** (Hessian Matrix Structure for Data-Level Inference). *Let $H(\boldsymbol{w}, \boldsymbol{\sigma})$ be the Hessian matrix of data-level $\boldsymbol{\sigma}$-weighted objective (5) with respect to $\boldsymbol{w}$, i.e., $H(\boldsymbol{w}, \boldsymbol{\sigma}) = \frac{\partial^2 \mathcal{L}(\boldsymbol{w}, \boldsymbol{\sigma})}{\partial \boldsymbol{w} \partial \boldsymbol{w}^\top}$, then we have*

$$
H(\boldsymbol{w}, \boldsymbol{\sigma}) = \begin{pmatrix} H_{1,1} & \cdots & 0 & H_{1,K+1} \\ \vdots & \ddots & \vdots & \vdots \\ 0 & \cdots & H_{K,K} & H_{K,K+1} \\ H_{K+1,1} & \cdots & H_{K+1,K} & H_{K+1,K+1} \end{pmatrix},
$$

*where*

$$
H_{kk} = \sum_{i=1}^{n_k} \sigma_{ki} \frac{\partial^2 \ell_{ki}(\theta_k, \gamma)}{\partial \theta_k \partial \theta_k^\top} + \frac{\partial^2 \Omega_k(\theta_k, \gamma)}{\partial \theta_k \partial \theta_k^\top} \in \mathbb{R}^{d_k \times d_k} \quad for\ 1 \le k \le K,
$$

$$
H_{kl} = 0 \in \mathbb{R}^{d_k \times d_l} \quad for\ 1 \le k, l \le K\ and\ k \ne l,
$$

$$
H_{K+1,k}^\top = H_{k,K+1} = \sum_{i=1}^{n_k} \sigma_{ki} \frac{\partial^2 \ell_{ki}(\theta_k, \gamma)}{\partial \theta_k \partial \gamma^\top} + \frac{\partial^2 \Omega_k(\theta_k, \gamma)}{\partial \theta_k \partial \gamma^\top} \in \mathbb{R}^{d_k \times p} \quad for\ 1 \le k \le K,
$$

$$
H_{K+1,K+1} = \sum_{k=1}^{K} \sum_{i=1}^{n_k} \sigma_{ki} \frac{\partial^2 \ell_{ki}(\theta_k, \gamma)}{\partial \gamma \partial \gamma^\top} + \sum_{k=1}^{K} \frac{\partial^2 \Omega_k(\theta_k, \gamma)}{\partial \gamma \partial \gamma^\top} \in \mathbb{R}^{p \times p}.
$$

**Lemma A.2** (Hessian Matrix Structure for Task-Level Inference). *Let $H(\boldsymbol{w}, \boldsymbol{\sigma})$ be the Hessian matrix of task-level $\boldsymbol{\sigma}$-weighted objective (11) with respect to $\boldsymbol{w}$, then*

$$
H(\boldsymbol{w}, \boldsymbol{\sigma}) = \begin{pmatrix} H_{1,1} & \cdots & 0 & H_{1,K+1} \\ \vdots & \ddots & \vdots & \vdots \\ 0 & \cdots & H_{K,K} & H_{K,K+1} \\ H_{K+1,1} & \cdots & H_{K+1,K} & H_{K+1,K+1} \end{pmatrix},
$$

*where*

$$
H_{kk} = \sigma_k \left[ \sum_{i=1}^{n_k} \frac{\partial^2 \ell_{ki}(\theta_k, \gamma)}{\partial \theta_k \partial \theta_k^\top} + \frac{\partial^2 \Omega_k(\theta_k, \gamma)}{\partial \theta_k \partial \theta_k^\top} \right] \in \mathbb{R}^{d_k \times d_k} \quad for\ 1 \le k \le K,
$$

$$
H_{kl} = 0 \in \mathbb{R}^{d_k \times d_l} \quad for\ 1 \le k, l \le K\ and\ k \ne l,
$$

$$
H_{K+1,k}^\top = H_{k,K+1} = \sigma_k \left[ \sum_{i=1}^{n_k} \frac{\partial^2 \ell_{ki}(\theta_k, \gamma)}{\partial \theta_k \partial \gamma^\top} + \frac{\partial^2 \Omega_k(\theta_k, \gamma)}{\partial \theta_k \partial \gamma^\top} \right] \in \mathbb{R}^{d_k \times p} \quad for\ 1 \le k \le K,
$$

$$
H_{K+1,K+1} = \sum_{k=1}^{K} \sigma_k \left[ \sum_{i=1}^{n_k} \frac{\partial^2 \ell_{ki}(\theta_k, \gamma)}{\partial \gamma \partial \gamma^\top} + \frac{\partial^2 \Omega_k(\theta_k, \gamma)}{\partial \gamma \partial \gamma^\top} \right] \in \mathbb{R}^{p \times p}.
$$

**Lemma A.3** (Influence Scores for Data-Level Analysis). *Assume that the objective $\mathcal{L}(\boldsymbol{w}, \boldsymbol{\sigma})$ is twice differentiable and strictly convex in $\boldsymbol{w}$. Then, $\hat{\boldsymbol{w}}(\boldsymbol{\sigma}) = \arg\min_{\boldsymbol{w}} \mathcal{L}(\boldsymbol{w}, \boldsymbol{\sigma})$ satisfies $\frac{\partial \mathcal{L}(\hat{\boldsymbol{w}}(\boldsymbol{\sigma}), \boldsymbol{\sigma})}{\partial \boldsymbol{w}} = 0$. Moreover, we have:*

$$
\frac{\partial \hat{\boldsymbol{w}}(\boldsymbol{\sigma})}{\partial \sigma_{ki}} = -H(\hat{\boldsymbol{w}}(\boldsymbol{\sigma}), \boldsymbol{\sigma})^{-1} \left( 0, \cdots, 0, \underbrace{\frac{\partial \ell_{ki}}{\partial \theta_k^\top}}_{k\text{-th block}}, 0, \cdots, 0, \underbrace{\frac{\partial \ell_{ki}}{\partial \gamma^\top}}_{(K+1)\text{-th block}} \right)^\top,
$$

*where $H(\boldsymbol{w}, \boldsymbol{\sigma}) \in \mathbb{R}^{(\sum_{k=1}^{K} d_k + p) \times (\sum_{k=1}^{K} d_k + p)}$ is the Hessian matrix of $\mathcal{L}(\boldsymbol{w}, \boldsymbol{\sigma})$ with respect to $\boldsymbol{w}$.*

*Proof.* The result is obtained by applying the classical influence function framework as outlined in Koh & Liang (2017).  □

**Lemma A.4** (Influence Scores for Task-Level Analysis). *Assume that the objective $\mathcal{L}(\boldsymbol{w}, \boldsymbol{\sigma})$ is twice differentiable and strictly convex in $\boldsymbol{w}$. Then, the optimal solution $\hat{\boldsymbol{w}}(\boldsymbol{\sigma}) = \arg\min_{\boldsymbol{w}} \mathcal{L}(\boldsymbol{w}, \boldsymbol{\sigma})$*

*satisfies* $\frac{\partial \mathcal{L}(\hat{\boldsymbol{w}}(\boldsymbol{\sigma}), \boldsymbol{\sigma})}{\partial \boldsymbol{w}} = 0$. *Furthermore, we have:*

$$\frac{\partial \hat{\boldsymbol{w}}(\boldsymbol{\sigma})}{\partial \sigma_k} = -H(\hat{\boldsymbol{w}}(\boldsymbol{\sigma}), \boldsymbol{\sigma})^{-1} \left( 0, \cdots, 0, \underbrace{\sum_{i=1}^{n_k} \frac{\partial \ell_{ki}}{\partial \theta_k} + \frac{\partial \Omega_k}{\partial \theta_k}}_{k\text{-th block}}, 0, \cdots, 0, \underbrace{\sum_{i=1}^{n_k} \frac{\partial \ell_{ki}}{\partial \gamma} + \frac{\partial \Omega_k}{\partial \gamma}}_{(K+1)\text{-th block}} \right)^{\top},$$

*where $H(\boldsymbol{w}, \boldsymbol{\sigma}) \in \mathbb{R}^{(\sum_{k=1}^K d_k + p) \times (\sum_{k=1}^K d_k + p)}$ is the Hessian matrix of $\mathcal{L}(\boldsymbol{w}, \boldsymbol{\sigma})$ with respect to $\boldsymbol{w}$.*

*Proof.* The result is obtained by applying the classical influence function framework as outlined in Koh & Liang (2017). $\qquad\square$

The following two lemmas provide tools for verifying the invertibility of the Hessian matrix and calculating its inverse.

**Lemma A.5** (Invertibility of Hessian). *If $H_{kk}$ is invertible for $1 \le k \le K$, define*

$$N := H_{K+1, K+1} - \sum_{k=1}^K H_{K+1, k} H_{kk}^{-1} H_{k, K+1} \in \mathbb{R}^{p \times p}. \tag{A.1}$$

*If $N$ is also invertible, then $H$ is invertible.*

*Proof.* The proof is in Section B. $\qquad\square$

**Lemma A.6** (Hessian Inverse). *Let $\left[ H^{-1} \right]_{k,l}$ denote the $(k, l)$ block of the inverse Hessian $H(\boldsymbol{w}, \boldsymbol{\sigma})^{-1}$. Then for $1 \le k, l \le K$,*

$$
\begin{aligned}
\left[ H^{-1} \right]_{k,l} &= \mathbf{1}(k = l) \cdot H_{kk}^{-1} + H_{kk}^{-1} H_{k, K+1} N^{-1} H_{K+1, l} H_{ll}^{-1}, && \text{for } 1 \le k, l \le K, \\
\left[ H^{-1} \right]_{k, K+1} &= -H_{kk}^{-1} H_{k, K+1} N^{-1}, && \text{for } 1 \le k \le K, \\
\left[ H^{-1} \right]_{K+1, K+1} &= N^{-1}.
\end{aligned}
$$

*Proof.* The proof is in Section B. $\qquad\square$

# B  PROOF

*Proof of Lemma A.5 and Lemma A.6.* Denote

$$H = \begin{pmatrix} A & B \\ C & D \end{pmatrix},$$

where $A = \begin{pmatrix} H_{11} & & 0 \\ & \ddots & \\ 0 & & H_{KK} \end{pmatrix} \in \mathbb{R}^{(\sum_{k=1}^K n_k) \times (\sum_{k=1}^K n_k)}$, $B = C^{\top} = \begin{pmatrix} H_{1, K+1} \\ \vdots \\ H_{K, K+1} \end{pmatrix} \in$

$\mathbb{R}^{(\sum_{k=1}^K n_k) \times p}$, and $D = H_{K+1, K+1} \in \mathbb{R}^{p \times p}$. Under the conditions, the matrices $H_{kk}$ for $1 \le k \le K$ are invertible. Note that $A$ is a diagonal block matrix. It is also invertible and its inverse is given by

$$A^{-1} = \begin{pmatrix} H_{11}^{-1} & & \\ & \ddots & \\ & & H_{KK}^{-1} \end{pmatrix}.$$

In addition, under the conditions, $D - CA^{-1}B = H_{K+1, K+1} - \sum_{k=1}^K H_{K+1, k} H_{kk}^{-1} H_{k, K+1} = N$ is invertible. Using the inverse formula for block matrix, we have

$$H^{-1} = \begin{pmatrix} \left( A - BD^{-1}C \right)^{-1} & -A^{-1} B \left( D - CA^{-1}B \right)^{-1} \\ -D^{-1} C \left( A - BD^{-1}C \right)^{-1} & \left( D - CA^{-1}B \right)^{-1} \end{pmatrix}, \tag{B.2}$$

where the upper left block is equivalent to

$$\left( A - BD^{-1}C \right)^{-1} = A^{-1} + A^{-1} B \left( D - CA^{-1}B \right)^{-1} CA^{-1},$$

by using the Woodbury matrix identity. Further, by expanding the RHS of Equation (B.2) in terms of the blocks in $H$, we can get the block-wise expression of $H^{-1}$. In particular, for $1 \le k, l \le K$,

$$\left[H^{-1}\right]_{k,l} \equiv \left[\left(A - BD^{-1}C\right)^{-1}\right]_{k,l} = 1(k = l) \cdot H_{kk}^{-1} + \left[A^{-1}B\left(D - CA^{-1}B\right)^{-1}CA^{-1}\right]_{kl}$$

$$= 1(k = l) \cdot H_{kk}^{-1} + H_{kk}^{-1}H_{k,K+1} \cdot N^{-1} \cdot H_{K+1,l}H_{ll}^{-1}.$$

Further, for $1 \le k \le K$,

$$\left[H^{-1}\right]_{k,K+1} = \left[H^{-1}\right]_{K+1,k}^{\top} = H_{kk}^{-1}H_{k,K+1}N^{-1},$$

and

$$\left[H^{-1}\right]_{K+1,K+1} = N^{-1}.$$

$\square$

## C  EXPERIMENTS

### C.1  ADDITIONAL RESULTS AND DETAILS ON MTIF'S APPROXIMATION TO LOO AND LOTO

#### C.1.1  SYNTHETIC DATASET

The synthetic dataset for multi-task linear regression is generated with $m = 10$ tasks, where each dataset contains $n = 200$ samples $(x_{ji}, y_{ji})$, split into training and test sets. The input vectors $x_{ji}$ are independently sampled from a normal distribution $\mathcal{N}(0, I_d)$ with dimensionality $d = 50$. The response $y_{ji}$ is generated using a linear model $y_{ji} = x_{ji}^{\top}\theta_j^{\star} + \epsilon_{ji}$, where $\epsilon_{ji} \sim \mathcal{N}(0, 1)$ is independent noise.

The coefficient vectors $\theta_j^{\star}$ for task $j$ are generated by starting with a common vector $\beta^{\star} = 2e_1$ (where $e_1$ is a unit vector) and adding random perturbations $\delta_j$, sampled from a sphere with norm $\delta$. For a fraction $\epsilon m$ of the tasks, $\theta_j^{\star}$ is replaced with independent random vectors. This parameterization introduces variability in task similarity, with $\delta$ controlling the perturbation magnitude and $\epsilon$ determining the fraction of unrelated tasks. For more details, we refer readers to Duan & Wang (2023).

To explore different task similarity scenarios, we generate datasets under varying $\delta$ and $\epsilon$ values. The datasets are randomly divided into training, validation, and test sets with an 1:1:1 ratio.

**Leave-One-Out (LOO).** Figures 2 and 3 present results on the synthetic dataset for each task selected as the target task under varying $\delta$ and $\epsilon$. Additionally, Figures 4 and 5 show results when data to be removed originates from tasks different from those in the main text. In all cases, the strong linear relationship between MTIF scores and the gold-standard LOO scores is preserved, demonstrating that MTIF aligns well with LOO.

**Leave-One-Task-Out (LOTO).** Tables 4, 5, 6, 7, 8, and 9 present results under various combinations of $\delta$ and $\epsilon$. The high correlation scores across these settings further demonstrate that MTIF strongly aligns with LOTO, even in neural network settings.

#### C.1.2  HUMAN ACTIVITY RECOGNITION (HAR) DATASET

The Human Activity Recognition (HAR) dataset (Anguita et al., 2013) was constructed from recordings of 30 volunteers performing various daily activities while carrying smartphones equipped with inertial sensors on their waist. Each participant contributed an average of 343.3 samples, ranging from 281 to 409. Each sample corresponds to one of six activities: walking, walking upstairs, walking downstairs, sitting, standing, or lying.

The feature vectors for each sample are 561-dimensional, capturing information from both the time and frequency domains, and are reduced to 100 dimensions using Principal Component Analysis (PCA). To frame the dataset as a multitask learning problem, we treat each volunteer as a separate task. The goal is formulated as a multi-task logistic regression problem to classify whether a participant is sitting or engaged in any other activity. For each task, 10% of the data is randomly selected for testing, another 10% for validation, and the remaining data is used for training.

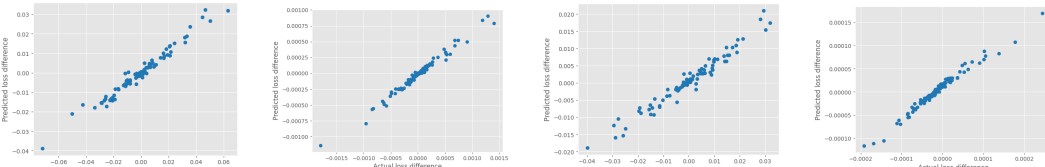

Figure 2: LOO experiments on linear regression. The x-axis is the actual loss difference obtained by LOO retraining, and the y-axis is the predicted loss difference calculated by MTIF. The first two figures from the left show within-task and between-task LOO (in order) results with $\delta = 0.4$ and $\epsilon = 0$, while the other two figures present within-task and between-task results (in order) with $\delta = 0.4$ and $\epsilon = 0.2$.

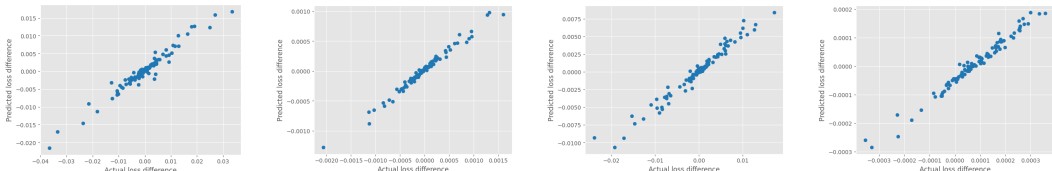

Figure 3: LOO experiments on linear regression. The x-axis is the actual loss difference obtained by LOO retraining, and the y-axis is the predicted loss difference calculated by MTIF. The first two figures from the left show within-task and between-task LOO (in order) results with $\delta = 0.8$ and $\epsilon = 0$, while the other two figures present within-task and between-task results (in order) with $\delta = 0.8$ and $\epsilon = 0.2$.

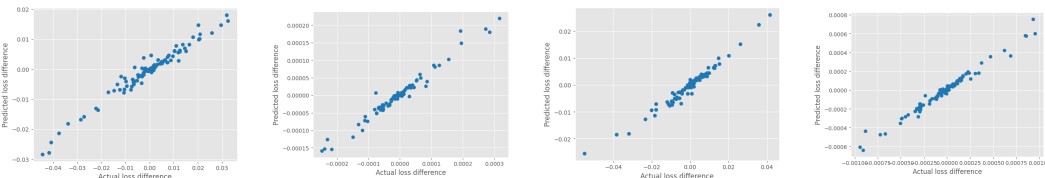

Figure 4: LOO experiments on linear regression. The x-axis is the actual loss difference obtained by LOO retraining, and the y-axis is the predicted loss difference calculated by MTIF. The first two figures from the left show within-task and between-task LOO (in order) results with deleted data from task 1, while the other two figures present within-task and between-task results (in order) with deleted data from task 2.

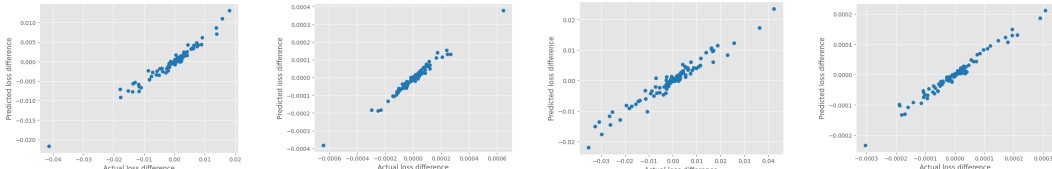

Figure 5: LOO experiments on linear regression. The x-axis is the actual loss difference obtained by LOO retraining, and the y-axis is the predicted loss difference calculated by MTIF. The first two figures from the left show within-task and between-task LOO (in order) results with deleted data from task 3, while the other two figures present within-task and between-task results (in order) with deleted data from task 5.

| Task 1 | Task 2 | Task 3 | Task 4 | Task 5 |
| --- | --- | --- | --- | --- |
| $0.84 \pm 0.05$ | $0.72 \pm 0.05$ | $0.74 \pm 0.11$ | $0.81 \pm 0.05$ | $0.71 \pm 0.09$ |

| Task 6 | Task 7 | Task 8 | Task 9 | Task 10 |
| --- | --- | --- | --- | --- |
| $0.74 \pm 0.04$ | $0.74 \pm 0.07$ | $0.84 \pm 0.03$ | $0.74 \pm 0.03$ | $0.65 \pm 0.07$ |

Table 4: The average Spearman correlation coefficients over 5 random seeds on the synthetic dataset. $\delta = 1.0$ and $\epsilon = 0.2$

| Task 1 | Task 2 | Task 3 | Task 4 | Task 5 |
| --- | --- | --- | --- | --- |
| $0.75 \pm 0.07$ | $0.67 \pm 0.06$ | $0.81 \pm 0.03$ | $0.70 \pm 0.05$ | $0.60 \pm 0.10$ |

| Task 6 | Task 7 | Task 8 | Task 9 | Task 10 |
| --- | --- | --- | --- | --- |
| $0.39 \pm 0.13$ | $0.66 \pm 0.06$ | $0.75 \pm 0.03$ | $0.71 \pm 0.05$ | $0.61 \pm 0.03$ |

Table 5: The average Spearman correlation coefficients over 5 random seeds on the synthetic dataset. $\delta = 1.0$ and $\epsilon = 0$.

| Task 1 | Task 2 | Task 3 | Task 4 | Task 5 |
| --- | --- | --- | --- | --- |
| $0.84 \pm 0.04$ | $0.67 \pm 0.07$ | $0.69 \pm 0.12$ | $0.77 \pm 0.05$ | $0.71 \pm 0.05$ |

| Task 6 | Task 7 | Task 8 | Task 9 | Task 10 |
| --- | --- | --- | --- | --- |
| $0.73 \pm 0.07$ | $0.65 \pm 0.06$ | $0.77 \pm 0.05$ | $0.69 \pm 0.05$ | $0.56 \pm 0.11$ |

Table 6: The average Spearman correlation coefficients over 5 random seeds on the synthetic dataset. $\delta = 0.6$ and $\epsilon = 0.2$.

| Task 1 | Task 2 | Task 3 | Task 4 | Task 5 |
| --- | --- | --- | --- | --- |
| $0.77 \pm 0.05$ | $0.56 \pm 0.09$ | $0.69 \pm 0.07$ | $0.63 \pm 0.06$ | $0.57 \pm 0.13$ |

| Task 6 | Task 7 | Task 8 | Task 9 | Task 10 |
| --- | --- | --- | --- | --- |
| $0.38 \pm 0.16$ | $0.62 \pm 0.04$ | $0.72 \pm 0.03$ | $0.65 \pm 0.04$ | $0.46 \pm 0.09$ |

Table 7: The average Spearman correlation coefficients over 5 random seeds on the synthetic dataset. $\delta = 0.6$ and $\epsilon = 0$.

| Task 1 | Task 2 | Task 3 | Task 4 | Task 5 |
| --- | --- | --- | --- | --- |
| $0.79 \pm 0.05$ | $0.62 \pm 0.06$ | $0.56 \pm 0.13$ | $0.73 \pm 0.05$ | $0.64 \pm 0.07$ |

| Task 6 | Task 7 | Task 8 | Task 9 | Task 10 |
| --- | --- | --- | --- | --- |
| $0.67 \pm 0.08$ | $0.52 \pm 0.05$ | $0.70 \pm 0.04$ | $0.65 \pm 0.04$ | $0.56 \pm 0.09$ |

Table 8: The average Spearman correlation coefficients over 5 random seeds on the synthetic dataset. $\delta = 0.4$ and $\epsilon = 0.2$.

| Task 1 | Task 2 | Task 3 | Task 4 | Task 5 |
| --- | --- | --- | --- | --- |
| $0.67 \pm 0.08$ | $0.52 \pm 0.10$ | $0.56 \pm 0.09$ | $0.64 \pm 0.06$ | $0.54 \pm 0.15$ |

| Task 6 | Task 7 | Task 8 | Task 9 | Task 10 |
| --- | --- | --- | --- | --- |
| $0.42 \pm 0.16$ | $0.52 \pm 0.08$ | $0.65 \pm 0.05$ | $0.56 \pm 0.04$ | $0.38 \pm 0.12$ |

Table 9: The average Spearman correlation coefficients over 5 random seeds on the synthetic dataset. $\delta = 0.4$ and $\epsilon = 0$.

### C.1.3 Neural Network Experimental Settings

We further evaluate MTIF in neural network settings on the CelebA dataset (Liu et al., 2015). CelebA is a large-scale face image dataset annotated with 40 attributes and widely used in the multitask learning (MTL) literature (Fifty et al., 2021). Following the setup in Fifty et al. (2021), we select 9 attributes as tasks for our experiments, modeling each task as a binary classification problem. The dataset is pre-partitioned into training, validation, and test sets. For our experiments, we sample a subset of 1000 examples per task from each partition to construct our training, validation, and test sets.

The neural network model is trained for 200 epochs using a StepLR learning rate scheduler with a step size of 100 and $\gamma = 0.5$. Optimization is performed using cross-entropy loss and the Adam optimizer (Kingma & Ba, 2017) without weight decay, ensuring the regularization term is zero.

### C.2 Results on Task Relatedness

We incorporate two gradient-based baselines into our task-relatedness experiments for both linear regression and neural network settings: Cosine Similarity (Azorin et al., 2023) and TAG (Fifty et al.,

2021). Following the procedure outlined in Section 5.1, we evaluate task relatedness by designating one task as the target task, ranking the most influential tasks relative to it as calculated by MTIF, Cosine Similarity, or TAG, and computing the ranking correlation coefficient with the ground-truth Leave-One-Task-Out (LOTO) scores. A higher correlation coefficient indicates better alignment with the LOTO scores, with values ranging from -1 (completely reversed alignment) to 1 (perfect alignment), and 0 representing random ranking.

| Task | Task 1 | Task 2 | Task 3 | Task 4 | Task 5 |
|---|---|---|---|---|---|
| MTIF | $0.84 \pm 0.05$ | $0.72 \pm 0.05$ | $0.74 \pm 0.11$ | $0.81 \pm 0.05$ | $0.71 \pm 0.09$ |
| TAG | $0.57 \pm 0.03$ | $0.63 \pm 0.07$ | $0.49 \pm 0.11$ | $0.56 \pm 0.05$ | $0.69 \pm 0.04$ |
| Cosine | $0.52 \pm 0.04$ | $0.48 \pm 0.07$ | $0.39 \pm 0.12$ | $0.47 \pm 0.09$ | $0.58 \pm 0.06$ |
| Task | Task 6 | Task 7 | Task 8 | Task 9 | Task 10 |
| MTIF | $0.74 \pm 0.04$ | $0.74 \pm 0.07$ | $0.84 \pm 0.03$ | $0.74 \pm 0.03$ | $0.65 \pm 0.07$ |
| TAG | $0.55 \pm 0.12$ | $0.42 \pm 0.06$ | $0.44 \pm 0.24$ | $0.66 \pm 0.08$ | $0.61 \pm 0.07$ |
| Cosine | $0.47 \pm 0.12$ | $0.34 \pm 0.05$ | $0.40 \pm 0.22$ | $0.62 \pm 0.09$ | $0.51 \pm 0.08$ |

Table 10: The average Spearman correlation coefficients over 5 random seeds on the synthetic dataset for MTIF, TAG, and Cosine across 10 tasks.

| | Task 1 | Task 2 | Task 3 | Task 4 | Task 5 | Task 6 |
|---|---|---|---|---|---|---|
| MTIF | $0.87 \pm 0.02$ | $0.90 \pm 0.02$ | $0.88 \pm 0.01$ | $0.91 \pm 0.03$ | $0.91 \pm 0.01$ | $0.90 \pm 0.02$ |
| TAG | $0.26 \pm 0.13$ | $0.42 \pm 0.11$ | $0.55 \pm 0.09$ | $0.22 \pm 0.07$ | $0.60 \pm 0.07$ | $0.55 \pm 0.08$ |
| Cosine | $0.31 \pm 0.11$ | $0.40 \pm 0.11$ | $0.57 \pm 0.08$ | $0.20 \pm 0.09$ | $0.61 \pm 0.06$ | $0.57 \pm 0.08$ |
| | Task 7 | Task 8 | Task 9 | Task 10 | Task 11 | Task 12 |
| MTIF | $0.90 \pm 0.01$ | $0.88 \pm 0.02$ | $0.92 \pm 0.01$ | $0.91 \pm 0.02$ | $0.89 \pm 0.02$ | $0.86 \pm 0.01$ |
| TAG | $0.49 \pm 0.12$ | $0.31 \pm 0.12$ | $0.24 \pm 0.01$ | $0.33 \pm 0.02$ | $0.43 \pm 0.03$ | $0.21 \pm 0.02$ |
| Cosine | $0.46 \pm 0.11$ | $0.31 \pm 0.14$ | $0.26 \pm 0.03$ | $0.34 \pm 0.01$ | $0.46 \pm 0.04$ | $0.18 \pm 0.11$ |
| | Task 13 | Task 14 | Task 15 | Task 16 | Task 17 | Task 18 |
| MTIF | $0.90 \pm 0.02$ | $0.93 \pm 0.05$ | $0.84 \pm 0.01$ | $0.87 \pm 0.05$ | $0.89 \pm 0.02$ | $0.82 \pm 0.02$ |
| TAG | $0.54 \pm 0.03$ | $0.57 \pm 0.03$ | $0.43 \pm 0.02$ | $0.48 \pm 0.03$ | $0.64 \pm 0.05$ | $0.44 \pm 0.02$ |
| Cosine | $0.53 \pm 0.10$ | $0.58 \pm 0.10$ | $0.48 \pm 0.04$ | $0.49 \pm 0.11$ | $0.66 \pm 0.05$ | $0.46 \pm 0.07$ |
| | Task 19 | Task 20 | Task 21 | Task 22 | Task 23 | Task 24 |
| MTIF | $0.85 \pm 0.02$ | $0.91 \pm 0.02$ | $0.93 \pm 0.02$ | $0.80 \pm 0.01$ | $0.80 \pm 0.02$ | $0.82 \pm 0.05$ |
| TAG | $0.44 \pm 0.03$ | $0.46 \pm 0.02$ | $0.84 \pm 0.02$ | $0.52 \pm 0.07$ | $0.13 \pm 0.03$ | $0.38 \pm 0.07$ |
| Cosine | $0.48 \pm 0.05$ | $0.47 \pm 0.07$ | $0.84 \pm 0.10$ | $0.53 \pm 0.08$ | $0.16 \pm 0.12$ | $0.45 \pm 0.10$ |
| | Task 25 | Task 26 | Task 27 | Task 28 | Task 29 | Task 30 |
| MTIF | $0.89 \pm 0.02$ | $0.81 \pm 0.03$ | $0.82 \pm 0.03$ | $0.89 \pm 0.01$ | $0.92 \pm 0.03$ | $0.86 \pm 0.03$ |
| TAG | $0.56 \pm 0.04$ | $0.14 \pm 0.11$ | $0.41 \pm 0.10$ | $0.14 \pm 0.11$ | $0.72 \pm 0.04$ | $0.41 \pm 0.11$ |
| Cosine | $0.60 \pm 0.04$ | $0.18 \pm 0.12$ | $0.46 \pm 0.10$ | $0.15 \pm 0.10$ | $0.74 \pm 0.11$ | $0.46 \pm 0.10$ |

Table 11: The average Spearman correlation coefficients over 5 random seeds on HAR dataset for MTIF, TAG, and Cosine across 30 tasks.

| | Task 1 | Task 2 | Task 3 | Task 4 | Task 5 |
|---|---|---|---|---|---|
| MTIF | $0.23 \pm 0.08$ | $0.44 \pm 0.19$ | $0.25 \pm 0.11$ | $0.36 \pm 0.12$ | $0.17 \pm 0.13$ |
| TAG | $-0.10 \pm 0.13$ | $-0.10 \pm 0.14$ | $0.09 \pm 0.06$ | $0.40 \pm 0.08$ | $0.00 \pm 0.12$ |
| Cosine | $0.12 \pm 0.18$ | $0.08 \pm 0.15$ | $0.08 \pm 0.07$ | $0.37 \pm 0.08$ | $-0.10 \pm 0.13$ |
| | Task 6 | Task 7 | Task 8 | Task 9 | |
| MTIF | $0.35 \pm 0.08$ | $0.25 \pm 0.07$ | $0.11 \pm 0.09$ | $0.18 \pm 0.12$ | |
| TAG | $-0.42 \pm 0.08$ | $-0.26 \pm 0.17$ | $0.06 \pm 0.13$ | $0.16 \pm 0.16$ | |
| Cosine | $-0.25 \pm 0.12$ | $-0.25 \pm 0.14$ | $-0.01 \pm 0.16$ | $0.05 \pm 0.12$ | |

Table 12: The average Spearman correlation coefficients over 5 random seeds on CelebA dataset for MTIF, TAG, and Cosine across 9 tasks.

The results in Tables 10, 11, and 12 show that our proposed MTIF method consistently outperforms the baselines across all scenarios. For the synthetic and HAR datasets, all methods achieve positive correlation scores across tasks, but MTIF consistently achieves the highest scores, often exceeding 0.7 for most tasks. In the CelebA dataset, estimating task relatedness in neural network models proves to be more challenging. While MTIF maintains positive scores, the baselines perform close

to random, frequently yielding negative scores for many tasks. Although the baselines occasionally achieve slightly higher scores than MTIF on specific tasks, their performance is inconsistent. These findings underscore MTIF's reliability and superior ability to approximate task relatedness compared to the baselines.

## C.3 ADDITIONAL RESULTS ON DATA SELECTION

In this section, we present additional results on data selection. First, we evaluate the impact of MTIF-enabled data selection under relatively simpler linear models on the synthetic dataset and HAR dataset (Anguita et al., 2013). These experiments provide interpretable insights into the performance gains achieved through MTIF-enabled data selection. Second, we include results demonstrating the flexibility of MTIF-enabled data selection when combined with multitask architectures beyond the shared-bottom architecture on the CelebA dataset (Liu et al., 2015).

### C.3.1 SYNTHETIC DATASET

|  | Task 1 | Task 2 | Task 3 | Task 4 | Task 5 | Task 6 | Task 7 | Task 8 | Task 9 | Task 10 |
|---|---|---|---|---|---|---|---|---|---|---|
| Before DS | 0.304 | 0.316 | 0.442 | 0.269 | 0.303 | 0.335 | 0.420 | 0.322 | 0.458 | 0.291 |
| After DS | 0.291 | 0.303 | 0.418 | 0.252 | 0.284 | 0.324 | 0.412 | 0.300 | 0.436 | 0.284 |

Table 13: Average loss across all tasks on synthetic dataset before and after data selection.

For the synthetic dataset, a straightforward metric for model performance is the $l_2$ distance between the estimated parameters $\hat{\theta}_j$ and the true parameters $\theta_j^*$ for each task $j$. As shown in Table 13, the $l_2$ distance between the estimated and true parameters decreases for most tasks after applying MTIF-enabled data selection, demonstrating the efficacy of our method.

### C.3.2 HAR DATASET

For the real-world HAR dataset, we investigate the benefits of MTIF-enabled data selection through a label-flipping experiment. We introduce adversarial noise by randomly flipping 5% of the labels in a small subset of the training data. During data selection, an equivalent portion of the data is removed using the proposed MTIF method. Task-wise classification errors are reported in Table 14. The results show that the average classification error decreases significantly after applying data selection, highlighting the robustness of MTIF in mitigating the impact of noisy data.

|  | Vanilla | Clustered | Lowrank |
|---|---|---|---|
| Before DS | 0.020 | 0.018 | 0.017 |
| After DS | 0.017 | 0.012 | 0.011 |

Table 14: Average classification loss for each task with different multitask learning methods (Vanilla, Clustered, and Lowrank) proposed in Duan & Wang (2023) before and after data selection.

### C.3.3 DIFFERENT MODEL ARCHITECTURES

We also evaluate the impact of MTIF-enabled data selection across different multitask learning architectures, including CGC (Tang et al., 2020), MMoE (Ma et al., 2018), and DSelect_k (Hazimeh et al., 2021). For consistency, we fix the weighting strategy to equal weighting and vary the model architecture to train models before and after data selection. As our primary interest is in the effect of data selection, we do not perform hyperparameter tuning. The results in Table 15 indicate that applying MTIF-enabled data selection leads to significant performance improvements for most tasks across all model architectures.

### C.4 LISSA

We empirically demonstrate that as the number of tasks $K$ increases, LiSSA requires larger batch sizes for stochastic estimation to address issues arising from a high condition number. Using a synthetic dataset for a linear regression task, we show that increasing $K$ necessitates larger batch sizes to achieve stable convergence. In our experiments, we vary the batch size and observe the convergence rate by measuring the ratio of the $l_2$-distance between the ground-truth Hessian and the

|          | Task 1 | Task 2 | Task 3 | Task 4 | Task 5 | Task 6 | Task 7 | Task 8 | Task 9 | Average |
|----------|--------|--------|--------|--------|--------|--------|--------|--------|--------|---------|
| CGC      | 0.863  | 0.772  | 0.878  | 0.734  | 0.920  | 0.939  | 0.834  | 0.900  | 0.949  | 0.866   |
| CGC+DS   | 0.868  | 0.783  | 0.877  | 0.766  | 0.925  | 0.943  | 0.842  | 0.917  | 0.956  | 0.875   |
| DSelect_k | 0.855 | 0.786  | 0.862  | 0.758  | 0.927  | 0.947  | 0.850  | 0.913  | 0.950  | 0.872   |
| DSelect_k+DS | 0.868 | 0.787 | 0.867 | 0.775 | 0.933  | 0.951  | 0.856  | 0.924  | 0.954  | 0.880   |
| HPS      | 0.859  | 0.815  | 0.896  | 0.791  | 0.934  | 0.951  | 0.872  | 0.919  | 0.958  | 0.888   |
| HPS+DS   | 0.872  | 0.825  | 0.896  | 0.802  | 0.935  | 0.954  | 0.868  | 0.927  | 0.961  | 0.893   |
| MMoE     | 0.843  | 0.793  | 0.881  | 0.740  | 0.917  | 0.944  | 0.842  | 0.899  | 0.956  | 0.868   |
| MMoE+DS  | 0.868  | 0.793  | 0.887  | 0.766  | 0.929  | 0.949  | 0.867  | 0.926  | 0.959  | 0.883   |

Table 15: Performance comparison across different tasks and model architectures with and without data selection (DS). The average performance over 5 random seeds is reported in the last column.

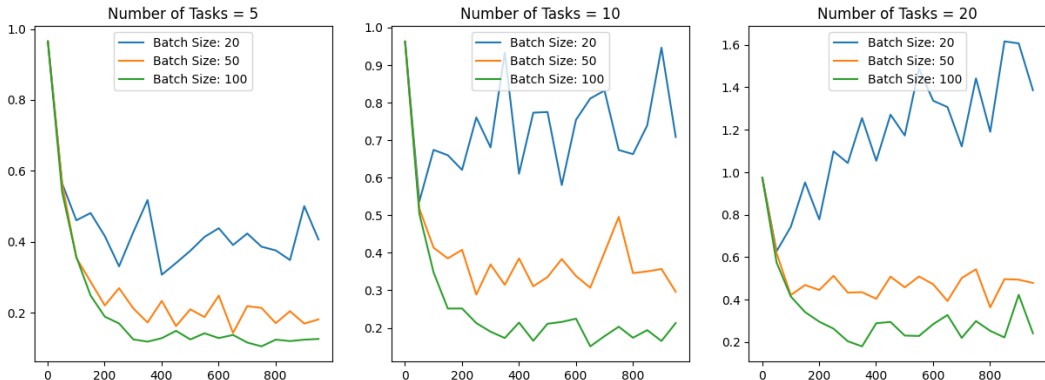

Figure 6: Convergence analysis of LiSSA-estimated Hessians across varying recursion depths and batch sizes. Each plot represents results for a different number of tasks, with three lines indicating the performance for different batch sizes. The y-axis shows the L2-distance ratio between the ground-truth Hessian and the LiSSA-estimated Hessian, while the x-axis denotes the recursion depth.

LiSSA-estimated Hessian to the norm of the ground-truth Hessian ($y$-axis). As shown in Figure 6, the need for larger batch sizes becomes increasingly evident as $K$ grows.

We further analyze the impact of the condition number in stochastic Hessian estimation by varying the batch size and the number of tasks. The condition number is defined as the ratio of the largest eigenvalue to the smallest eigenvalue of the Hessian. A larger condition number indicates a more ill-conditioned Hessian. As shown in Table 16, the condition number increases with the number of tasks and decreases with larger batch sizes. Notably, the condition number becomes infinite when the batch size is 100 and the number of tasks is 20, indicating that some tasks lack samples in the Hessian calculation, resulting in at least one zero eigenvalue. This highlights the necessity of using larger batch sizes in LiSSA to maintain a well-conditioned Hessian, although this compromises efficiency.

|    | 100     | 150     | 200     | 250     | 300    | Full  |
|----|---------|---------|---------|---------|--------|-------|
| 5  | 1602.53 | 417.03  | 177.06  | 95.82   | 65.96  | 22.38 |
| 10 | 7395.75 | 1583.46 | 647.36  | 378.03  | 236.25 | 20.47 |
| 20 | inf     | 8067.93 | 2959.49 | 1533.45 | 911.80 | 18.34 |

Table 16: Average condition number of batched Hessians computed over 100 random seeds. Each column represents the selected batch size, with "Full" indicating the full Hessian. Each task contains 100 samples, and each row corresponds to a different number of tasks. Notably, the condition number becomes infinite when the batch size is 100 and the number of tasks is 20.

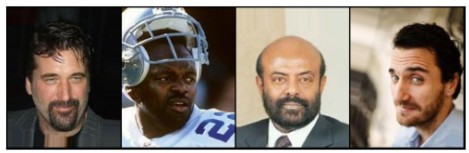 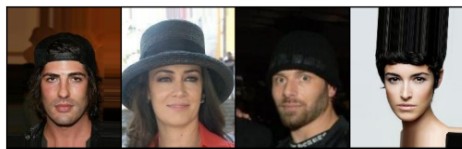

Figure 7: The images on the left represent four samples from the task "Mustache" that negatively influence the task "No Beard." They are labeled positive for "Mustache" but negative for "No Beard." On the right, there are four samples from the task "Wearing Hat" that negatively influence the task "Black Hair." They are labeled positive for "Wearing Hat" but negative for "Black Hair."

## D  VISUALIZATION OF MOST NEGATIVE SAMPLES

In this section, we demonstrate how MTIF can provide interpretable insights into task relatedness. Using the CelebA dataset, we visualize some of the most negative samples between specific task pairs in Figure 7.

On the left side of Figure 7, we show samples from the task "Mustache" that negatively influence the task "No Beard." Intuitively, these tasks are related, as individuals with a mustache often have a beard. However, the images depicted are negative samples for the "Mustache" task, yet the individuals clearly have beards. Such samples could potentially confuse the model, as the visual cues contradict the task label.

Similarly, the images on the right side of Figure 7 depict individuals wearing a black hat but labeled as not having black hair. This labeling might occur because the individual's natural hair color is not black or because their hair is obscured by the hat, though this distinction is not obvious from the images. The model may misinterpret the presence of a black hat as indicating black hair.

These examples illustrate that MTIF can identify samples from one task that negatively influence another, providing interpretable insights into task relationships and potential sources of confusion for the model.

