# OpenReview forum: "Data Attribution for Multitask Learning"
_ICLR.cc/2025/Conference — Submitted to ICLR 2025_

### Official Review · Reviewer_MsCP · 2024-10-19

**Soundness:** 2
**Presentation:** 4
**Contribution:** 3
**Rating:** 5
**Confidence:** 3

**Summary:**

This paper extends traditional data attribution techniques from single-task learning to multi-task learning (MTL), addressing the interference caused by cross-task parameter sharing on sample gradients. By analyzing task relatedness, the paper introduces a novel approach to data-level analysis, offering a novel perspective for enhancing MTL.

**Strengths:**

1. The paper provides a novel perspective for modeling task relatedness in multi-task learning by investigating the influence of individual samples on other tasks. This fine-grained analysis helps elucidate the interference caused by cross-task sharing.

2. The data selection method, realized through data-level influence analysis in Section 5.2.2, presents a unique and valuable application for MTL, distinguishing this work from existing approaches.

**Weaknesses:**

1. The main contribution of the paper lies in applying influence functions to MTL in the form of Leave-One-Out (LOO) and Leave-One-Task-Out (LOTO) analyses, a conceptually straightforward extension. The key distinction from single-task learning is the influence of the task-sharing quantity $\Omega_k$ in Eq. 5. However, under the assumption of a convex combination of tasks, this influence is relatively easy to analyze and does not provide a particularly challenging technical novelty.

2. The work lacks direct baselines, resulting in insufficient comparison. For instance, in Tables 1 and 2, the experiments are only compared with the ground truth LOTO scores, making it difficult to assess the performance of the method quantitatively. This is especially concerning given the unsatisfactory correlation coefficient results in Table 2.

3. While this may be a somewhat harsh critique, it is worth noting that traditional MTL methods are rapidly being supplanted by more advanced models. Recent work in MTL since 2022 increasingly leverages cutting-edge techniques such as multi-modal models (e.g., CLIP) or fine-tuning large language models (LLMs) via LoRA (both included in the Related Work section). In contrast, the experimental setup and models employed here appear overly simplistic, serving more as illustrative case studies than as a demonstration of broader applicability or superiority in real-world scenarios.

**Questions:**

1. Would it be feasible to adapt existing single-task attribution algorithms to provide additional baselines for the experiments presented in Sections 5.1.2 and 5.2.1? This would improve the rigor of the empirical evaluation and provide more meaningful comparisons.

2. In Section 5.2.2, the authors report significant improvements in MTL performance by removing certain negative samples. However, is this approach entirely justified? Many recent MTL methods address conflicting gradients by adjusting only the conflicting components (e.g., gradient surgery and its variants), preserving parts of the conflicting samples that are beneficial to their respective tasks. Could the proposed algorithm be extended to provide insights or improvements to these gradient-based optimization techniques?

---

> ### Author Response · Authors · 2024-11-27
>
> We thank Reviewer MsCP for taking the time to review our paper and for their constructive feedback. Please find below our point-to-point response:
>
> > **Contribution**:
>
> We acknowledge that applying influence functions (IF) to multitask learning (MTL) may appear conceptually similar with IF for single-task learning (STL). However, our contribution lies in two significant aspects:
> 1. Adapting single-task IF to the MTL setting requires a new framework due to the unique parameter structure in MTL. Specifically, MTL involves both shared and task-specific parameters, and test data predictions in MTL are tied to only a submodel within the overall framework. Addressing these complexities necessitated rethinking the application of influence functions in this context.
> 2. Our method introduces a natural way to estimate task-relatedness and address negative transfer, two critical challenges in MTL. This contribution is particularly relevant to the MTL literature, as understanding and mitigating negative transfer has significant implications for improving MTL performance.
>
> Furthermore, motivated by the reviewer’s question, we further investigated the potential adaptation of efficient approximate IF methods developed for STL to MTL, and showed that it is non-trivial. Specifically, we examined two common Hessian inverse approximation techniques used in STL IF settings: EK-FAC and LiSSA. Our derivations provide insights into their applicability in MTL settings. The updated paper includes a detailed discussion (see Section 4.2), which we summarize below:
> - **EK-FAC**: This method approximates the Hessian using a blockwise diagonal matrix, which ignores off-diagonal interactions between shared and task-specific parameters. While computationally efficient, this approximation can lead to the loss of significant contributions when computing influence scores, particularly in soft parameter-sharing models where inter-task interactions play a critical role.
> - **LiSSA**: This method approximates the *inverse-Hessian-vector-product* using an iterative algorithm that supports mini-batch gradients. In MTL settings, however, the empirical Hessian for a data point has a unique structure due to parameter sharing, with non-zero entries restricted to specific sub-blocks. This structure often results in the mini-batch empirical Hessian being ill-posed, characterized by a high condition number, which poses challenges for achieving convergence and numerical stability. In this revision, we ran additional experiments to assess the applicability of LiSSA and added the results to the Appendix. Our empirical results suggest that, as the number of tasks increases, LiSSA requires progressively larger batch sizes to stabilize the stochastic approximation. This scaling significantly raises the computational costs for large-scale MTL problems. Adapting popular methods like LiSSA to address challenges arising from the unique Hessian structure in MTL settings requires nontrivial efforts and would be a valuable direction for future work.
>
> > **Experiment Baselines**:
>
> We have incorporated two gradient-based baselines, TAG [1] and Cosine Similarity [2], into our task-relatedness experiments for both linear regression and neural networks. These baselines are methods for measuring task relatedness in the MTL literature. Each baseline method provides a score of task relatedness for each pair of tasks. We evaluate these methods in terms of the correlation between their scores and the oracle task relatedness obtained from brute-force LOTO retraining as detailed in our paper.
>
>
> The results, as shown below, clearly demonstrate that our proposed MTIF method outperforms these baselines. Specifically, MTIF achieves consistently higher correlation coefficients with oracle influence estimates, underscoring its superior effectiveness in quantifying task-relatedness. We have included these new results in Appendix Section C.2, and we list below for your convenience.

---

> ### Author Response · Authors · 2024-11-27
>
> (continued) Results for synthetic dataset:
> | Method / Task | Task 1          | Task 2          | Task 3          | Task 4          | Task 5          |
> |---------------|-----------------|-----------------|-----------------|-----------------|-----------------|
> | Ours          | 0.84 ± 0.05    | 0.72 ± 0.05    | 0.74 ± 0.11    | 0.81 ± 0.05    | 0.71 ± 0.09    |
> | TAG           | 0.57 ± 0.03    | 0.63 ± 0.07    | 0.49 ± 0.11    | 0.56 ± 0.05    | 0.69 ± 0.04    |
> | Cosine        | 0.52 ± 0.04    | 0.48 ± 0.07    | 0.39 ± 0.12    | 0.47 ± 0.09    | 0.58 ± 0.06    |
>
> | Method / Task | Task 6          | Task 7          | Task 8          | Task 9          | Task 10         |
> |---------------|-----------------|-----------------|-----------------|-----------------|-----------------|
> | Ours          | 0.74 ± 0.04    | 0.74 ± 0.07    | 0.84 ± 0.03    | 0.74 ± 0.03    | 0.65 ± 0.07    |
> | TAG           | 0.55 ± 0.12    | 0.42 ± 0.06    | 0.44 ± 0.24    | 0.66 ± 0.08    | 0.61 ± 0.07    |
> | Cosine        | 0.47 ± 0.12    | 0.34 ± 0.05    | 0.40 ± 0.22    | 0.62 ± 0.09    | 0.51 ± 0.08    |
>
> Results for HAR dataset:
> | Method / Task | Task 1       | Task 2       | Task 3       | Task 4       | Task 5       | Task 6       |
> |---------------|--------------|--------------|--------------|--------------|--------------|--------------|
> | Ours          | 0.87 ± 0.02 | 0.90 ± 0.02 | 0.88 ± 0.01 | 0.91 ± 0.03 | 0.91 ± 0.01 | 0.90 ± 0.02 |
> | TAG           | 0.26 ± 0.13 | 0.42 ± 0.11 | 0.55 ± 0.09 | 0.22 ± 0.07 | 0.60 ± 0.07 | 0.55 ± 0.08 |
> | Cosine        | 0.31 ± 0.11 | 0.40 ± 0.11 | 0.57 ± 0.08 | 0.20 ± 0.09 | 0.61 ± 0.06 | 0.57 ± 0.08 |
>
> | Method / Task | Task 7       | Task 8       | Task 9       | Task 10      | Task 11      | Task 12      |
> |---------------|--------------|--------------|--------------|--------------|--------------|--------------|
> | Ours          | 0.90 ± 0.01 | 0.88 ± 0.02 | 0.92 ± 0.01 | 0.91 ± 0.02 | 0.89 ± 0.02 | 0.86 ± 0.01 |
> | TAG           | 0.49 ± 0.12 | 0.31 ± 0.12 | 0.24 ± 0.01 | 0.33 ± 0.02 | 0.43 ± 0.03 | 0.21 ± 0.02 |
> | Cosine        | 0.46 ± 0.11 | 0.31 ± 0.14 | 0.26 ± 0.03 | 0.34 ± 0.01 | 0.46 ± 0.04 | 0.18 ± 0.11 |
>
> | Method / Task | Task 13      | Task 14      | Task 15      | Task 16      | Task 17      | Task 18      |
> |---------------|--------------|--------------|--------------|--------------|--------------|--------------|
> | Ours          | 0.90 ± 0.02 | 0.93 ± 0.05 | 0.84 ± 0.01 | 0.87 ± 0.05 | 0.89 ± 0.02 | 0.82 ± 0.02 |
> | TAG           | 0.54 ± 0.03 | 0.57 ± 0.03 | 0.43 ± 0.02 | 0.48 ± 0.03 | 0.64 ± 0.05 | 0.44 ± 0.02 |
> | Cosine        | 0.53 ± 0.10 | 0.58 ± 0.10 | 0.48 ± 0.04 | 0.49 ± 0.11 | 0.66 ± 0.05 | 0.46 ± 0.07 |
>
> | Method / Task | Task 19      | Task 20      | Task 21      | Task 22      | Task 23      | Task 24      |
> |---------------|--------------|--------------|--------------|--------------|--------------|--------------|
> | Ours          | 0.85 ± 0.02 | 0.91 ± 0.02 | 0.93 ± 0.02 | 0.80 ± 0.01 | 0.80 ± 0.02 | 0.82 ± 0.05 |
> | TAG           | 0.44 ± 0.03 | 0.46 ± 0.02 | 0.84 ± 0.02 | 0.52 ± 0.07 | 0.13 ± 0.03 | 0.38 ± 0.07 |
> | Cosine        | 0.48 ± 0.05 | 0.47 ± 0.07 | 0.84 ± 0.10 | 0.53 ± 0.08 | 0.16 ± 0.12 | 0.45 ± 0.10 |
>
> | Method / Task | Task 25      | Task 26      | Task 27      | Task 28      | Task 29      | Task 30      |
> |---------------|--------------|--------------|--------------|--------------|--------------|--------------|
> | Ours          | 0.89 ± 0.02 | 0.81 ± 0.03 | 0.82 ± 0.03 | 0.89 ± 0.01 | 0.92 ± 0.03 | 0.86 ± 0.03 |
> | TAG           | 0.56 ± 0.04 | 0.14 ± 0.11 | 0.41 ± 0.10 | 0.14 ± 0.11 | 0.72 ± 0.04 | 0.41 ± 0.11 |
> | Cosine        | 0.60 ± 0.04 | 0.18 ± 0.12 | 0.46 ± 0.10 | 0.15 ± 0.10 | 0.74 ± 0.11 | 0.46 ± 0.10 |
>
> Results for CelebA dataset:
>
> | Method / Task | Task 1       | Task 2       | Task 3       | Task 4       | Task 5       |
> |---------------|--------------|--------------|--------------|--------------|--------------|
> | Ours          | 0.23 ± 0.08 | 0.44 ± 0.19 | 0.25 ± 0.11 | 0.36 ± 0.12 | 0.17 ± 0.13 |
> | TAG           | -0.10 ± 0.13 | -0.10 ± 0.14 | 0.09 ± 0.06 | 0.40 ± 0.08 | 0.00 ± 0.12 |
> | Cosine        | 0.12 ± 0.18 | 0.08 ± 0.15 | 0.08 ± 0.07 | 0.37 ± 0.08 | -0.10 ± 0.13 |
>
> | Method / Task | Task 6       | Task 7       | Task 8       | Task 9       |
> |---------------|--------------|--------------|--------------|--------------|
> | Ours          | 0.35 ± 0.08 | 0.25 ± 0.07 | 0.11 ± 0.09 | 0.18 ± 0.12 |
> | TAG           | -0.42 ± 0.08 | -0.26 ± 0.17 | 0.06 ± 0.13 | 0.16 ± 0.16 |
> | Cosine        | -0.25 ± 0.12 | -0.25 ± 0.14 | -0.01 ± 0.16 | 0.05 ± 0.12 |

---

> ### Author Response · Authors · 2024-11-27
>
> > **More Complex Models**:
>
> Thank you for suggesting the exploration of more advanced models, such as CLIP and LoRA. While this paper does not include experiments on such models, we would like to emphasize the following points:
> 1. Complex models also exhibit parameter-sharing structures, and our derivations remain relevant and insightful in these contexts. STL-based influence functions have already been generalized to complex models [3]. Therefore, we believe extending our method to these architectures is both feasible and an exciting direction for future work.
> 2. In many real-world applications that have strict requirements for serving time latency, simpler MTL models are still widely used due to their computational efficiency. Examples include recommender systems [4] and autonomous driving [5].
> 3. Certain scenarios, particularly those requiring interpretability, favor simpler models over complex architectures. This is especially important in domains like healthcare and finance [6,7], where understanding model predictions is crucial. As a result, simpler MTL models continue to hold significant academic interest (e.g., the linear models used in our experiments come from Duan and Wang (2023) published in Annals of Statistics [8]).
>
> > **Single-Task Method Baselines**:
>
> 1. The primary contribution of our work is adapting STL-based IF to the MTL setting, which necessitates a new framework. This adaptation involves addressing the unique challenges posed by task-specific and shared parameter structures in MTL. That's the focus of our paper.
> 2. For the experimental baselines, we have additionally included gradient-based task-relatedness measures from the MTL literature to provide further points of comparison and enhance the rigor of our evaluation.
>
>
> > **Relationship to Gradient-Based Optimization Techniques**:
>
> Our method and gradient-based optimization techniques are orthogonal approaches. While gradient-based techniques adjust conflicting gradients to preserve beneficial components during training, our data selection method focuses on identifying and mitigating the influence of negative samples via data selection. Notably, our data selection method can be combined with gradient-based techniques to achieve improved performance, as indicated by our experimental results.
>
> We sincerely appreciate your insightful feedback, which has allowed us to improve both the clarity and robustness of our work. Thank you again for your thoughtful comments!
>
> [1] Fifty, Chris, et al. "Efficiently identifying task groupings for multi-task learning." Advances in Neural Information Processing Systems 34 (2021): 27503-27516.
>
> [2] Azorin, Raphaël, et al. "" It's a Match!"--A Benchmark of Task Affinity Scores for Joint Learning." arXiv preprint arXiv:2301.02873 (2023).
>
> [3] Roger Grosse, Juhan Bae, Cem Anil, Nelson Elhage, Alex Tamkin, Amirhossein Tajdini, Benoit Steiner, Dustin Li, Esin Durmus, Ethan Perez, Evan Hubinger, Kamilė Lukošiūtė, Karina Nguyen, Nicholas Joseph, Sam McCandlish, Jared Kaplan, and Samuel R. Bowman. Studying large language model generalization with influence functions, 2023. URL https://arxiv.org/abs/2308.03296.
>
> [4] Zhao, Zhe, Lichan Hong, Li Wei, Jilin Chen, Aniruddh Nath, Shawn Andrews, Aditee Kumthekar, Maheswaran Sathiamoorthy, Xinyang Yi, and Ed Chi. "Recommending what video to watch next: a multitask ranking system." In Proceedings of the 13th ACM conference on recommender systems, pp. 43-51. 2019.
>
> [5] Liang, Xiwen, Yangxin Wu, Jianhua Han, Hang Xu, Chunjing Xu, and Xiaodan Liang. "Effective adaptation in multi-task co-training for unified autonomous driving." Advances in Neural Information Processing Systems 35 (2022): 19645-19658.
>
> [6] Parker Knight and Rui Duan. Multi-task learning with summary statistics. In A. Oh, T. Naumann, A. Globerson, K. Saenko, M. Hardt, and S. Levine (eds.), *Advances in Neural Information Processing Systems*, volume 36, pp. 54020–54031. Curran Associates, Inc., 2023.
>
> [7] Adel Javanmard, Jingwei Ji, and Renyuan Xu. Multi-task dynamic pricing in credit market with contextual information, 2024. URL https://arxiv.org/abs/2410.14839.
>
> [8] Yaqi Duan and Kaizheng Wang. Adaptive and robust multi-task learning. *The Annals of Statistics*, 51(5):2015 – 2039, 2023. doi: 10.1214/23-AOS2319. URL https://doi.org/10.1214/23-AOS2319.

---

### Official Review · Reviewer_yCgf · 2024-10-24

**Soundness:** 3
**Presentation:** 3
**Contribution:** 3
**Rating:** 6
**Confidence:** 3

**Summary:**

The authors extend influence functions to the multitask learning setting. Due to the fact that there are multiple sets of task-specific parameters, if done naively the Hessian of the loss with respect to the full set of parameters would be very large and thus difficult to invert. The authors circumvent this issue by using the block structure of the Hessian. On both synthetic and real datasets, the influence function values correlate strongly with the difference in validation loss attained by leave-one-out retraining. Moreover, removing the most problematic examples and retraining yields consistent performance improvements on the CelebA dataset.

**Strengths:**

* The work is well-motivated and tackles the important problem of data attribution in multitask learning.
* The novelty is significant, since influence functions have not yet been extended to this setting.
* The empirical side of the work is strong, since the task-level influence is correlated to the LOTO effect across three datasets.
* The authors take it a step further and demonstrate a concrete benefit of their work; performance on CelebA improves across the majority of tasks by removing the most problematic examples as measured by the task-level influence.

**Weaknesses:**

* There is no baseline to compare against in the results in Table 1 and 2, which makes the numbers difficult to interpret. I.e. is 21% correlation high or low? I understand that this is a new approach and there may not be existing methods to compare against. However there should be at least some kind of (perhaps trivial) baseline to compare against.

**Questions:**

* Include some kind of baseline for the correlation between task-level influence and LOTO effect.
* Can you report the data selection results (Table 3) on the synthetic and human action recognition datasets as well? It would be good to see that this yields consistent improves across multiple datasets, not just one.

---

> ### Author Response · Authors · 2024-11-27
>
> We thank Reviewer yCgf for taking the time to review our paper and for their constructive feedback. Please find below our point-to-point response:
> > **Experiment Baselines**:
>
> We have incorporated two gradient-based baselines, TAG [1] and Cosine Similarity [2], into our task-relatedness experiments for both linear regression and neural networks. These baselines are methods for measuring task relatedness in the MTL literature. Each baseline method provides a score of task relatedness for each pair of tasks. We evaluate these methods in terms of the correlation between their scores and the oracle task relatedness obtained from brute-force LOTO retraining as detailed in our paper.
>
>
> The results, as shown below, clearly demonstrate that our proposed MTIF method outperforms these baselines. Specifically, MTIF achieves consistently higher correlation coefficients with oracle influence estimates, underscoring its superior effectiveness in quantifying task-relatedness. We have included these new results in Appendix Section C.2, and we list below for your convenience.
>
> Results for synthetic dataset:
> | Method / Task | Task 1          | Task 2          | Task 3          | Task 4          | Task 5          |
> |---------------|-----------------|-----------------|-----------------|-----------------|-----------------|
> | Ours          | 0.84 ± 0.05    | 0.72 ± 0.05    | 0.74 ± 0.11    | 0.81 ± 0.05    | 0.71 ± 0.09    |
> | TAG           | 0.57 ± 0.03    | 0.63 ± 0.07    | 0.49 ± 0.11    | 0.56 ± 0.05    | 0.69 ± 0.04    |
> | Cosine        | 0.52 ± 0.04    | 0.48 ± 0.07    | 0.39 ± 0.12    | 0.47 ± 0.09    | 0.58 ± 0.06    |
>
> | Method / Task | Task 6          | Task 7          | Task 8          | Task 9          | Task 10         |
> |---------------|-----------------|-----------------|-----------------|-----------------|-----------------|
> | Ours          | 0.74 ± 0.04    | 0.74 ± 0.07    | 0.84 ± 0.03    | 0.74 ± 0.03    | 0.65 ± 0.07    |
> | TAG           | 0.55 ± 0.12    | 0.42 ± 0.06    | 0.44 ± 0.24    | 0.66 ± 0.08    | 0.61 ± 0.07    |
> | Cosine        | 0.47 ± 0.12    | 0.34 ± 0.05    | 0.40 ± 0.22    | 0.62 ± 0.09    | 0.51 ± 0.08    |
>
> Results for HAR dataset:
> | Method / Task | Task 1       | Task 2       | Task 3       | Task 4       | Task 5       | Task 6       |
> |---------------|--------------|--------------|--------------|--------------|--------------|--------------|
> | Ours          | 0.87 ± 0.02 | 0.90 ± 0.02 | 0.88 ± 0.01 | 0.91 ± 0.03 | 0.91 ± 0.01 | 0.90 ± 0.02 |
> | TAG           | 0.26 ± 0.13 | 0.42 ± 0.11 | 0.55 ± 0.09 | 0.22 ± 0.07 | 0.60 ± 0.07 | 0.55 ± 0.08 |
> | Cosine        | 0.31 ± 0.11 | 0.40 ± 0.11 | 0.57 ± 0.08 | 0.20 ± 0.09 | 0.61 ± 0.06 | 0.57 ± 0.08 |
>
> | Method / Task | Task 7       | Task 8       | Task 9       | Task 10      | Task 11      | Task 12      |
> |---------------|--------------|--------------|--------------|--------------|--------------|--------------|
> | Ours          | 0.90 ± 0.01 | 0.88 ± 0.02 | 0.92 ± 0.01 | 0.91 ± 0.02 | 0.89 ± 0.02 | 0.86 ± 0.01 |
> | TAG           | 0.49 ± 0.12 | 0.31 ± 0.12 | 0.24 ± 0.01 | 0.33 ± 0.02 | 0.43 ± 0.03 | 0.21 ± 0.02 |
> | Cosine        | 0.46 ± 0.11 | 0.31 ± 0.14 | 0.26 ± 0.03 | 0.34 ± 0.01 | 0.46 ± 0.04 | 0.18 ± 0.11 |
>
> | Method / Task | Task 13      | Task 14      | Task 15      | Task 16      | Task 17      | Task 18      |
> |---------------|--------------|--------------|--------------|--------------|--------------|--------------|
> | Ours          | 0.90 ± 0.02 | 0.93 ± 0.05 | 0.84 ± 0.01 | 0.87 ± 0.05 | 0.89 ± 0.02 | 0.82 ± 0.02 |
> | TAG           | 0.54 ± 0.03 | 0.57 ± 0.03 | 0.43 ± 0.02 | 0.48 ± 0.03 | 0.64 ± 0.05 | 0.44 ± 0.02 |
> | Cosine        | 0.53 ± 0.10 | 0.58 ± 0.10 | 0.48 ± 0.04 | 0.49 ± 0.11 | 0.66 ± 0.05 | 0.46 ± 0.07 |
>
> | Method / Task | Task 19      | Task 20      | Task 21      | Task 22      | Task 23      | Task 24      |
> |---------------|--------------|--------------|--------------|--------------|--------------|--------------|
> | Ours          | 0.85 ± 0.02 | 0.91 ± 0.02 | 0.93 ± 0.02 | 0.80 ± 0.01 | 0.80 ± 0.02 | 0.82 ± 0.05 |
> | TAG           | 0.44 ± 0.03 | 0.46 ± 0.02 | 0.84 ± 0.02 | 0.52 ± 0.07 | 0.13 ± 0.03 | 0.38 ± 0.07 |
> | Cosine        | 0.48 ± 0.05 | 0.47 ± 0.07 | 0.84 ± 0.10 | 0.53 ± 0.08 | 0.16 ± 0.12 | 0.45 ± 0.10 |
>
> | Method / Task | Task 25      | Task 26      | Task 27      | Task 28      | Task 29      | Task 30      |
> |---------------|--------------|--------------|--------------|--------------|--------------|--------------|
> | Ours          | 0.89 ± 0.02 | 0.81 ± 0.03 | 0.82 ± 0.03 | 0.89 ± 0.01 | 0.92 ± 0.03 | 0.86 ± 0.03 |
> | TAG           | 0.56 ± 0.04 | 0.14 ± 0.11 | 0.41 ± 0.10 | 0.14 ± 0.11 | 0.72 ± 0.04 | 0.41 ± 0.11 |
> | Cosine        | 0.60 ± 0.04 | 0.18 ± 0.12 | 0.46 ± 0.10 | 0.15 ± 0.10 | 0.74 ± 0.11 | 0.46 ± 0.10 |

---

> ### Author Response · Authors · 2024-11-27
>
> (Continued) Results for CelebA dataset:
>
> | Method / Task | Task 1       | Task 2       | Task 3       | Task 4       | Task 5       |
> |---------------|--------------|--------------|--------------|--------------|--------------|
> | Ours          | 0.23 ± 0.08 | 0.44 ± 0.19 | 0.25 ± 0.11 | 0.36 ± 0.12 | 0.17 ± 0.13 |
> | TAG           | -0.10 ± 0.13 | -0.10 ± 0.14 | 0.09 ± 0.06 | 0.40 ± 0.08 | 0.00 ± 0.12 |
> | Cosine        | 0.12 ± 0.18 | 0.08 ± 0.15 | 0.08 ± 0.07 | 0.37 ± 0.08 | -0.10 ± 0.13 |
>
> | Method / Task | Task 6       | Task 7       | Task 8       | Task 9       |
> |---------------|--------------|--------------|--------------|--------------|
> | Ours          | 0.35 ± 0.08 | 0.25 ± 0.07 | 0.11 ± 0.09 | 0.18 ± 0.12 |
> | TAG           | -0.42 ± 0.08 | -0.26 ± 0.17 | 0.06 ± 0.13 | 0.16 ± 0.16 |
> | Cosine        | -0.25 ± 0.12 | -0.25 ± 0.14 | -0.01 ± 0.16 | 0.05 ± 0.12 |
>
>
> > **Additional Data Selection Results**:
>
> We have also provided the data selection results for the synthetic dataset and the HAR dataset [1] below as well as in Appendix C.3. Overall, We can see significant improvement after applying data selection in each dataset.
>
> The data selection results for synthetic dataset (the error between the learned parameter and the ground-truth for each task, the lower the better):
> |            | Task 1 | Task 2 | Task 3 | Task 4 | Task 5 | Task 6 | Task 7 | Task 8 | Task 9 | Task 10 |
> |------------|--------|--------|--------|--------|--------|--------|--------|--------|--------|---------|
> | Before DS  | 0.304  | 0.316  | 0.442  | 0.269  | 0.303  | 0.335  | 0.420  | 0.322  | 0.458  | 0.291   |
> | After DS   | 0.291  | 0.303  | 0.418  | 0.252  | 0.284  | 0.324  | 0.412  | 0.300  | 0.436  | 0.284   |
>
> The data selection results* for HAR dataset (average test classification error over all tasks, the lower the better; Vanilla, Clustered, and Lowrank refer to 3 MTL models used in [2]):
> |               | Vanilla | Clustered | Lowrank |
> |---------------|---------|-----------|---------|
> | Before DS     | 0.020   | 0.018     | 0.017   |
> | After DS      | 0.017   | 0.012     | 0.011   |
>
> *We note that the HAR dataset is a rather simple dataset, and the test classification errors of the original MTL models trained on the full dataset without data selection are already around 0.01. Therefore, it is difficult to have any further improvement on this dataset. In this experiment, we added noise to a random 5% of the training data points, which makes the original MTL models trained on this noisy dataset have test errors around 0.02. As can be seen from the table above, our data selection method brings the performance back to a 0.01 level of test error.
>
>
> [1] Reyes-Ortiz, Jorge, et al. "Human Activity Recognition Using Smartphones." UCI Machine Learning Repository, 2013, https://doi.org/10.24432/C54S4K.
>
> [2] Duan, Yaqi, and Kaizheng Wang. "Adaptive and robust multi-task learning." The Annals of Statistics 51.5 (2023): 2015-2039.

---

### Official Review · Reviewer_edvo · 2024-10-29

**Soundness:** 3
**Presentation:** 3
**Contribution:** 2
**Rating:** 5
**Confidence:** 4

**Summary:**

This paper proposes the MultiTask Influence Function (MTIF), a novel data attribution method for multitask learning (MTL). By extending influence function-based approaches to MTL, the authors aim to quantify data and task importance within MTL frameworks, offering insights into task relatedness and the impact of individual data points on model performance. The method shows promise in improving model efficiency and interpretability within MTL.

**Strengths:**

1. This paper introduces data attribution to the field of multitask learning (MTL) for the first time, analyzing data importance across multiple levels. This is a meaningful contribution, as understanding the significance of individual data points in MTL could greatly enhance the interpretability and performance of MTL models. The proposed work provides an efficient approach to quantify task relatedness, which is essential in MTL since task interactions often drive overall performance improvements.

2. The authors effectively extend the influence function (IF) to MTL through the proposed MultiTask Influence Function (MTIF), enabling both data-level and task-level attribution analysis. By circumventing the computational expense of retraining models, MTIF approximates leave-one-out (LOO) and leave-one-task-out (LOTO) effects, demonstrating efficient computational performance with strong interpretability, as supported by the experimental results.

**Weaknesses:**

1. Many conclusions in this paper are based on the assumptions in Equation 1; however, a detailed derivation for this equation is missing. This absence may impact readers’ confidence in the method’s validity. I recommend including a thorough derivation of Equation 1 to clarify MTIF’s applicability in MTL scenarios.

2. The derivation of Equation 6 relies on partial derivatives with respect to \(\sigma\), which, in turn, depend on the local properties of the parameter \(\theta\). These local properties may change across different training stages, and the importance of specific samples and tasks may shift as training progresses. For example, while derivatives may tend toward zero after convergence, this is not necessarily the case in earlier training stages. I suggest discussing ways to address this dynamic nature to ensure that influence estimates remain stable throughout the training process.

3. The paper does not sufficiently detail the sample selection process (e.g., whether samples are chosen randomly or deterministically), which may lead to a significant influence from randomness. Providing more detailed descriptions of the experimental setup, including criteria for selecting tasks and samples and data partitioning methods, would enhance the reproducibility and reliability of the results.

4. The experiments predominantly use simpler datasets, with much of the analysis focusing on cases of local properties (e.g., \(\sigma=1\) or \(0\)). For more complex datasets, substantial variations between discrete values may emerge, potentially challenging the evaluation methods used. I recommend extending the experiments to more complex MTL tasks (such as CIFAR-100) to assess the generalizability of MTIF and verify its effectiveness in complex scenarios.

**Questions:**

See Weakness

---

> ### Author Response · Authors · 2024-11-27
>
> We thank Reviewer edvo for taking the time to review our paper and for their constructive feedback. Please find below our point-to-point response:
>
> > **Equation (1)**:
>
> Equation (1) is a known result from the prior work [1], which is why its derivation was omitted in the initial version of the paper. For a detailed derivation of Equation (1), we refer the reviewer to the Appendix A of Koh and Liang (2017) [1]. We have also included a footnote in our paper to refer the readers to [1].
>
> > **Equation (6)**:
>
> The derivation of Equation (6) relies on learned parameters and is unaffected by early training stages, as highlighted in [1]. While the summation of derivatives over all data points equals zero at the global optimum, individual derivatives remain non-zero. As a result, the influence scores calculated from these derivatives remain meaningful and do not converge to zero, ensuring their interpretability and utility throughout.
>
> > **Sample Selection**:
>
> For the HAR dataset, we partition the entire dataset randomly into training, validation, and test sets using an 8:1:1 ratio.
> For the synthetic dataset, after data generation, we similarly divide the dataset randomly into training, validation, and test sets with an 1:1:1 ratio.
> For the CelebA dataset, the data is pre-partitioned into training, validation, and test sets. From each partition, we sample a subset of size 1000 for each task to construct our corresponding training, validation, and test sets. We also randomly sample 9 attributes from all 40 attributes to model as 9 binary classification tasks.
>
> > **More Complex Dataset (such as CIFAR-100)**:
>
> We note that our experiments have included a fairly complex dataset, the CelebA dataset. The CelebA dataset was introduced at ICCV 2015 by [3], whereas the CIFAR-100 dataset suggested by the reviewer was introduced in 2009 by [4]. CelebA comprises over 200,000 celebrity images, each annotated with 40 binary attributes, covering a wide range of facial features and expressions. Successful predictions on CelebA data require capturing the nuanced facial features in the image, which is often considered more complex than the object classification task in CIFAR-100.
>
> Furthermore, CelebA has been widely used as a standard benchmark in the MTL literature [5], as it is natural to convert the annotated attributes into multiple tasks. It is less natural to convert CIFAR-100, a multi-class classification dataset, into an MTL benchmark.
>
> > **Generalization Beyond the Current Scope**:
>
> Existing literature suggests that single-task learning (STL) influence functions generalize effectively to more complex datasets and models [2]. By extension, it is theoretically plausible for multitask learning (MTL) influence functions to generalize similarly. However, extending our method to more complex datasets and architectures lies beyond the scope of the current paper. We aim to explore this direction in future work.
>
> We sincerely appreciate the thoughtful feedback, which has helped us improve the clarity and robustness of our work. Thank you again for your valuable comments!
>
> [1] Pang Wei Koh and Percy Liang. Understanding black-box predictions via influence functions. In Doina Precup and Yee Whye Teh (eds.), *Proceedings of the 34th International Conference on Machine Learning*, volume 70 of *Proceedings of Machine Learning Research*, pp. 1885–1894. PMLR, 06–11 Aug 2017. URL https://proceedings.mlr.press/v70/koh17a.html.
>
> [2] Roger Grosse, Juhan Bae, Cem Anil, Nelson Elhage, Alex Tamkin, Amirhossein Tajdini, Benoit Steiner, Dustin Li, Esin Durmus, Ethan Perez, Evan Hubinger, Kamilė Lukošiūtė, Karina Nguyen, Nicholas Joseph, Sam McCandlish, Jared Kaplan, and Samuel R. Bowman. Studying large language model generalization with influence functions, 2023. URL https://arxiv.org/abs/2308.03296.
>
> [3] Ziwei Liu, Ping Luo, Xiaogang Wang, and Xiaoou Tang. Deep learning face attributes in the wild. In *Proceedings of the IEEE International Conference on Computer Vision (ICCV)*, December 2015
>
> [4] Alex Krizhevsky. Learning multiple layers of features from tiny images. 2009.
>
> [5] Chris Fifty, Ehsan Amid, Zhe Zhao, Tianhe Yu, Rohan Anil, and Chelsea Finn. Effi-
> ciently identifying task groupings for multi-task learning. In M. Ranzato, A. Beygelz-
> imer, Y. Dauphin, P.S. Liang, and J. Wortman Vaughan (eds.), *Advances in Neural In-
> formation Processing Systems*, volume 34, pp. 27503–27516. Curran Associates, Inc.,
> 2021. URL https://proceedings.neurips.cc/paper_files/paper/2021/
> file/e77910ebb93b511588557806310f78f1-Paper.pdf.

---

### Official Review · Reviewer_QvTU · 2024-11-04

**Soundness:** 2
**Presentation:** 2
**Contribution:** 2
**Rating:** 6
**Confidence:** 4

**Summary:**

This paper focuses on data-attribution and task-attribution in multitask learning. The authors propose a novel multitask influence function that efficiently estimates the impact of removing data points or excluding tasks on specific target task predictions, offering interpretable data-level and task-level influence analysis.

**Strengths:**

- This paper targets a meaningful problem in machine learning. In practice, interpretable data-level and task-level influence analysis is crucial in multitask learning.

- The motivation is stated clearly. The authors argue that re-training-based data attribution methods greatly increase computation cost, and that existing IF-based methods in single-task learning face computational challenges from complex optimization objectives. To address these issues, the authors propose a multitask influence function (MTIF).
- The experiments on multiple benchmark datasets validate the effectiveness of the proposed method compared with the baseline models.

**Weaknesses:**

- **The key differences between single-task IF and the proposed multitask IF should be clarified.** According to my understanding,  it seems the authors apply the idea of the IF-function (Koh & Liang, 2017) to multitask learning, handling more complex differential operations.

- **More empirical results are necessary.** On one hand, since the authors claim that the proposed method can enhance computational efficiency in the motivation, the efficacy analysis (such as memory usage, training/inference time, flops) also is crucial.  On the other hand, it is not convincing that the authors just select re-weighting-based method as competitors. The empirical results of directly transferring IF-based methods in STL to MTL need further analysis.

-  In the related work, the authors overlook several work [1,2,3] on addressing the ”negative transfer“ issue in multi-task learning.
- The authors should provide citations for each method in Tab. 3.

[1] Generalized Block-Diagonal Structure Pursuit: Learning Soft Latent Task Assignment against Negative Transfer. NeurIPS 2019.

[2]Multi-Task Distillation: Towards Mitigating the Negative Transfer in Multi-Task Learning. ICIP 2021.

[3]Feature Decomposition for Reducing Negative Transfer: A Novel Multi-task Learning Method for Recommender System. AAAI 2023.

**Questions:**

Please see the above weakness.

---

> ### Author Response · Authors · 2024-11-27
>
> We thank Reviewer QvTU for taking the time to review our paper and for their constructive feedback. Please find below our point-to-point response:
>
> > **The key differences between single-task IF and the proposed multitask IF**:
>
> We appreciate your comment regarding the distinction between single-task IF and our proposed multitask IF. Conceptually, our multitask IF is similar to single-task IF, as it builds on the influence function framework introduced by [1]. However, our contribution lies in two significant aspects:
> 1. Adapting single-task IF to the multitask learning (MTL) setting requires a new framework due to the unique parameter structure in MTL. Specifically, MTL involves both shared and task-specific parameters, and test data predictions in MTL are tied to only a submodel within the overall framework. Addressing these complexities necessitated rethinking the application of influence functions in this context.
> 2. Our method introduces a natural way to estimate task-relatedness and address negative transfer, two critical challenges in MTL. This contribution is particularly relevant to the MTL literature, as understanding and mitigating negative transfer has significant implications for improving multitask learning performance.
>
> > **Computational Complexity Analysis**:
>
> Thank you for raising this important point. We have added a remark in the method section regarding computational complexity. Specifically, for exact Hessian matrix inverse computation, our method reduces the computational complexity from $\Omega\left(\left(\sum_{k=1}^K d_k + p\right)^w\right)$ to $\Omega\left(\sum_{k=1}^K d_k^w + p^w\right)$, where $w \approx 2.37$, $d_k$ are the dimension of task specific parameters for task $k$, and $p$ is the dimension of shared parameters. This reduction is achieved by decoupling shared and task-specific parameters in the optimization process. For a detailed discussion, please refer to the updated method section in the paper.
>
> **Empirical Approximation to the Hessian Inverse**:
>
> Motivated by your suggestion, we have expanded the discussion in the method section to address potential approximations for the Hessian inverse, focusing on two widely used approaches: EK-FAC and LiSSA. Our derivations provide insights into their applicability in MTL settings. The updated paper includes a detailed discussion, which we summarize below:
> - **EK-FAC**: This method approximates the Hessian using a blockwise diagonal matrix, which ignores off-diagonal interactions between shared and task-specific parameters. While computationally efficient, this approximation can lead to the loss of significant contributions when computing influence scores, particularly in soft parameter-sharing models where inter-task interactions play a critical role.
> - **LiSSA**: This method approximates the *inverse-Hessian-vector-product* using an iterative algorithm that supports mini-batch gradients. In MTL settings, however, the empirical Hessian for a data point has a unique structure due to parameter sharing, with non-zero entries restricted to specific sub-blocks. This structure often results in the mini-batch empirical Hessian being ill-posed, characterized by a high condition number, which poses challenges for achieving convergence and numerical stability. In this revision, we ran additional experiments to assess the applicability of LiSSA and added the results to the Appendix. Our empirical results suggest that, as the number of tasks increases, LiSSA requires progressively larger batch sizes to stabilize the stochastic approximation. This scaling significantly raises the computational costs for large-scale MTL problems. Adapting popular methods like LiSSA to address challenges arising from the unique Hessian structure in MTL settings requires nontrivial efforts and would be a valuable direction for future work.

---

> ### Author Response · Authors · 2024-11-27
>
> > **More baseline comprison**:
>
> We would like to first clarify that IF-based methods in STL cannot be directly applied to the MTL setting due to the unique parameter structure in MTL models, as detailed in the first point of our response.
>
> To address the reviewer’s concern about limited baselines presented in this paper, we have included the following two sets of additional experiments.
>
> **1) We showed that our data selection can combine with different architectures.**
>
> We report the data selection results with different multitask architectures, namely CGC [6], MMoE [4] and DSelect-k [5]. The results demonstrate that, after applying data selection with MTIF, the test accuracy of most tasks shows significant improvement across all model architectures. These results have also been included in the Appendix Section C.3.3, and we list below for your reference.
>
> | Methods / Models | Task 1 | Task 2 | Task 3 | Task 4 | Task 5 | Task 6 | Task 7 | Task 8 | Task 9 | Average |
> |------------------|--------|--------|--------|--------|--------|--------|--------|--------|--------|---------|
> | CGC              | 0.863  | 0.772  | 0.878  | 0.734  | 0.920  | 0.939  | 0.834  | 0.900  | 0.949  | 0.866   |
> | CGC+DS           | 0.868  | 0.783  | 0.877  | 0.766  | 0.925  | 0.943  | 0.842  | 0.917  | 0.956  | 0.875   |
> | DSelect-k        | 0.855  | 0.786  | 0.862  | 0.758  | 0.927  | 0.947  | 0.850  | 0.913  | 0.950  | 0.872   |
> | DSelect-k+DS     | 0.868  | 0.787  | 0.867  | 0.775  | 0.933  | 0.951  | 0.856  | 0.924  | 0.954  | 0.880   |
> | HPS              | 0.859  | 0.815  | 0.896  | 0.791  | 0.934  | 0.951  | 0.872  | 0.919  | 0.958  | 0.888   |
> | HPS+DS           | 0.872  | 0.825  | 0.896  | 0.802  | 0.935  | 0.954  | 0.868  | 0.927  | 0.961  | 0.893   |
> | MMoE             | 0.843  | 0.793  | 0.881  | 0.740  | 0.917  | 0.944  | 0.842  | 0.899  | 0.956  | 0.868   |
> | MMoE+DS          | 0.868  | 0.793  | 0.887  | 0.766  | 0.929  | 0.949  | 0.867  | 0.926  | 0.959  | 0.883   |
>
> **2) We compared our influence score with gradient-based task-relatedness measurements in MTL literature.**
>
> We have incorporated two gradient-based baselines, TAG [2] and Cosine Similarity [3], into our task-relatedness experiments for both linear regression and neural networks. These baselines are methods for measuring task relatedness in the MTL literature. Each baseline method provides a score of task relatedness for each pair of tasks. We evaluate these methods in terms of the correlation between their scores and the oracle task relatedness obtained from brute-force LOTO retraining as detailed in our paper.
>
> The results, as shown below, clearly demonstrate that our proposed MTIF method outperforms these baselines. Specifically, MTIF achieves consistently higher correlation coefficients with oracle influence estimates, underscoring its superior effectiveness in quantifying task-relatedness. We have included these new results in Appendix Section C.2, and we list below for your convenience.
>
> Results for synthetic dataset:
> | Method / Task | Task 1          | Task 2          | Task 3          | Task 4          | Task 5          |
> |---------------|-----------------|-----------------|-----------------|-----------------|-----------------|
> | Ours          | 0.84 ± 0.05    | 0.72 ± 0.05    | 0.74 ± 0.11    | 0.81 ± 0.05    | 0.71 ± 0.09    |
> | TAG           | 0.57 ± 0.03    | 0.63 ± 0.07    | 0.49 ± 0.11    | 0.56 ± 0.05    | 0.69 ± 0.04    |
> | Cosine        | 0.52 ± 0.04    | 0.48 ± 0.07    | 0.39 ± 0.12    | 0.47 ± 0.09    | 0.58 ± 0.06    |
>
> | Method / Task | Task 6          | Task 7          | Task 8          | Task 9          | Task 10         |
> |---------------|-----------------|-----------------|-----------------|-----------------|-----------------|
> | Ours          | 0.74 ± 0.04    | 0.74 ± 0.07    | 0.84 ± 0.03    | 0.74 ± 0.03    | 0.65 ± 0.07    |
> | TAG           | 0.55 ± 0.12    | 0.42 ± 0.06    | 0.44 ± 0.24    | 0.66 ± 0.08    | 0.61 ± 0.07    |
> | Cosine        | 0.47 ± 0.12    | 0.34 ± 0.05    | 0.40 ± 0.22    | 0.62 ± 0.09    | 0.51 ± 0.08    |

---

> ### Author Response · Authors · 2024-11-27
>
> (continued) Results for HAR dataset:
> | Method / Task | Task 1       | Task 2       | Task 3       | Task 4       | Task 5       | Task 6       |
> |---------------|--------------|--------------|--------------|--------------|--------------|--------------|
> | Ours          | 0.87 ± 0.02 | 0.90 ± 0.02 | 0.88 ± 0.01 | 0.91 ± 0.03 | 0.91 ± 0.01 | 0.90 ± 0.02 |
> | TAG           | 0.26 ± 0.13 | 0.42 ± 0.11 | 0.55 ± 0.09 | 0.22 ± 0.07 | 0.60 ± 0.07 | 0.55 ± 0.08 |
> | Cosine        | 0.31 ± 0.11 | 0.40 ± 0.11 | 0.57 ± 0.08 | 0.20 ± 0.09 | 0.61 ± 0.06 | 0.57 ± 0.08 |
>
> | Method / Task | Task 7       | Task 8       | Task 9       | Task 10      | Task 11      | Task 12      |
> |---------------|--------------|--------------|--------------|--------------|--------------|--------------|
> | Ours          | 0.90 ± 0.01 | 0.88 ± 0.02 | 0.92 ± 0.01 | 0.91 ± 0.02 | 0.89 ± 0.02 | 0.86 ± 0.01 |
> | TAG           | 0.49 ± 0.12 | 0.31 ± 0.12 | 0.24 ± 0.01 | 0.33 ± 0.02 | 0.43 ± 0.03 | 0.21 ± 0.02 |
> | Cosine        | 0.46 ± 0.11 | 0.31 ± 0.14 | 0.26 ± 0.03 | 0.34 ± 0.01 | 0.46 ± 0.04 | 0.18 ± 0.11 |
>
> | Method / Task | Task 13      | Task 14      | Task 15      | Task 16      | Task 17      | Task 18      |
> |---------------|--------------|--------------|--------------|--------------|--------------|--------------|
> | Ours          | 0.90 ± 0.02 | 0.93 ± 0.05 | 0.84 ± 0.01 | 0.87 ± 0.05 | 0.89 ± 0.02 | 0.82 ± 0.02 |
> | TAG           | 0.54 ± 0.03 | 0.57 ± 0.03 | 0.43 ± 0.02 | 0.48 ± 0.03 | 0.64 ± 0.05 | 0.44 ± 0.02 |
> | Cosine        | 0.53 ± 0.10 | 0.58 ± 0.10 | 0.48 ± 0.04 | 0.49 ± 0.11 | 0.66 ± 0.05 | 0.46 ± 0.07 |
>
> | Method / Task | Task 19      | Task 20      | Task 21      | Task 22      | Task 23      | Task 24      |
> |---------------|--------------|--------------|--------------|--------------|--------------|--------------|
> | Ours          | 0.85 ± 0.02 | 0.91 ± 0.02 | 0.93 ± 0.02 | 0.80 ± 0.01 | 0.80 ± 0.02 | 0.82 ± 0.05 |
> | TAG           | 0.44 ± 0.03 | 0.46 ± 0.02 | 0.84 ± 0.02 | 0.52 ± 0.07 | 0.13 ± 0.03 | 0.38 ± 0.07 |
> | Cosine        | 0.48 ± 0.05 | 0.47 ± 0.07 | 0.84 ± 0.10 | 0.53 ± 0.08 | 0.16 ± 0.12 | 0.45 ± 0.10 |
>
> | Method / Task | Task 25      | Task 26      | Task 27      | Task 28      | Task 29      | Task 30      |
> |---------------|--------------|--------------|--------------|--------------|--------------|--------------|
> | Ours          | 0.89 ± 0.02 | 0.81 ± 0.03 | 0.82 ± 0.03 | 0.89 ± 0.01 | 0.92 ± 0.03 | 0.86 ± 0.03 |
> | TAG           | 0.56 ± 0.04 | 0.14 ± 0.11 | 0.41 ± 0.10 | 0.14 ± 0.11 | 0.72 ± 0.04 | 0.41 ± 0.11 |
> | Cosine        | 0.60 ± 0.04 | 0.18 ± 0.12 | 0.46 ± 0.10 | 0.15 ± 0.10 | 0.74 ± 0.11 | 0.46 ± 0.10 |
>
> Results for CelebA dataset:
>
> | Method / Task | Task 1       | Task 2       | Task 3       | Task 4       | Task 5       |
> |---------------|--------------|--------------|--------------|--------------|--------------|
> | Ours          | 0.23 ± 0.08 | 0.44 ± 0.19 | 0.25 ± 0.11 | 0.36 ± 0.12 | 0.17 ± 0.13 |
> | TAG           | -0.10 ± 0.13 | -0.10 ± 0.14 | 0.09 ± 0.06 | 0.40 ± 0.08 | 0.00 ± 0.12 |
> | Cosine        | 0.12 ± 0.18 | 0.08 ± 0.15 | 0.08 ± 0.07 | 0.37 ± 0.08 | -0.10 ± 0.13 |
>
> | Method / Task | Task 6       | Task 7       | Task 8       | Task 9       |
> |---------------|--------------|--------------|--------------|--------------|
> | Ours          | 0.35 ± 0.08 | 0.25 ± 0.07 | 0.11 ± 0.09 | 0.18 ± 0.12 |
> | TAG           | -0.42 ± 0.08 | -0.26 ± 0.17 | 0.06 ± 0.13 | 0.16 ± 0.16 |
> | Cosine        | -0.25 ± 0.12 | -0.25 ± 0.14 | -0.01 ± 0.16 | 0.05 ± 0.12 |
>
> > **Negative Transfer Literature**:
>
> Thank you for pointing out these valuable works on addressing negative transfer in multitask learning. While these approaches tackle the issue from different perspectives, they provide important context for understanding our contributions. We have now included citations and discussions of these works in the related work section of our paper to provide a more comprehensive review of the literature.

---

> ### Author Response · Authors · 2024-11-27
>
> > **Citations for Methods in Table 3**:
>
> Thank you for catching this oversight. We have added the appropriate citations for each method listed in Table 3 in the revised version of the paper.
>
> We sincerely appreciate your constructive comments, which have significantly improved the clarity and scope of our work. Thank you again for your thoughtful feedback!
>
>
> [1] Pang Wei Koh and Percy Liang. Understanding black-box predictions via influence functions. In Doina Precup and Yee Whye Teh (eds.), *Proceedings of the 34th International Conference on Machine Learning*, volume 70 of *Proceedings of Machine Learning Research*, pp. 1885–1894. PMLR, 06–11 Aug 2017. URL https://proceedings.mlr.press/v70/koh17a.html.
>
> [2] Fifty, Chris, et al. "Efficiently identifying task groupings for multi-task learning." Advances in Neural Information Processing Systems 34 (2021): 27503-27516.
>
> [3] Azorin, Raphaël, et al. "" It's a Match!"--A Benchmark of Task Affinity Scores for Joint Learning." arXiv preprint arXiv:2301.02873 (2023).
>
> [4] Ma, Jiaqi, et al. "Modeling task relationships in multi-task learning with multi-gate mixture-of-experts." Proceedings of the 24th ACM SIGKDD international conference on knowledge discovery & data mining. 2018.
>
> [5] Hazimeh, Hussein, et al. "Dselect-k: Differentiable selection in the mixture of experts with applications to multi-task learning." Advances in Neural Information Processing Systems 34 (2021): 29335-29347.
>
> [6] Tang, Hongyan, et al. "Progressive layered extraction (ple): A novel multi-task learning (mtl) model for personalized recommendations." Proceedings of the 14th ACM Conference on Recommender Systems. 2020.

---

### Official Review · Reviewer_4o4x · 2024-11-04

**Soundness:** 3
**Presentation:** 3
**Contribution:** 3
**Rating:** 6
**Confidence:** 4

**Summary:**

This paper proposes the MultiTask Influence Function (MTIF) by extending data attribution techniques for single-task learning (STL) to the context of multitask learning (MTL). The authors identify new challenges in data attribution in MTL which stems from task interdependencies and the need to balance shared and task-specific parameters. The proposed MTIF method approximates the impact of individual data points or entire tasks on other tasks’ performance, enabling both data-level and task-level influence analysis without the need for retraining. The authors validate MTIF’s effectiveness in approximating data-level and task-level influences through experiments on linear models and shallow neural networks, by demonstrating positive correlations with an oracle influence estimation from extensive computations.

**Strengths:**

**An important extension of data attribution problem and method for STL to the context of MTL:** A key contribution of this work is its formulation of data attribution specifically for MTL, highlighting the need to quantify data influence not just within tasks, but across multiple related tasks that share parameters.

**Clear presentation and justification of the proposed method:** The authors describe MTIF’s construction and rationale with clarity, particularly regarding its approach to handling shared and task-specific parameters in MTL. By leveraging the influence function (IF) approach, which uses first-order approximations to model parameter changes, MTIF mitigates the computational burden of retraining, which is commonly associated with data attribution methods.

**Preliminary yet convincing experimental results:** The experiments clearly demonstrate MTIF’s ability to approximate leave-one-out (LOO) and leave-one-task-out (LOTO) influences and to be employed in the downstream task after estimating data influences, although the setups are simplistic.

**Weaknesses:**

**No computational complexity analysis:** Although the major challenge of data attribution in MTL is computational complexity, the paper lacks explicit theoretical or empirical analysis of MTIF’s computational complexity. Given that influence functions and Hessian computations are generally costly, a complexity analysis would have been useful in understanding the method’s scalability, particularly for large-scale MTL applications.

**High computational cost of calculation of Hessian:** Despite MTIF’s efficiency improvements, Hessian computations remain costly, which could hinder the method’s application to high-dimensional models or a large number of tasks. The paper could benefit from discussing any further optimizations or Hessian approximations to address this issue.

**Limited experimental scope:** The experiments are conducted on simple models, including linear models and shallow neural networks, which may not fully showcase MTIF’s potential in realistic MTL scenarios. Evaluation on more sophisticated architectures, such as deep neural networks or transformer-based models, and larger datasets would provide a more comprehensive assessment.

**Lack of comparison with existing baselines:** The paper does not compare MTIF to simple methods for estimating task relatedness, such as cosine similarity of gradients, which are often used in MTL, although they are tailored for only estimating task-level influences. Additionally, incorporating heuristics based on gradient similarity could serve as promising baselines even for data-level influences, e.g., cosine similarity between the gradient of an individual data point and the average gradient.

**Questions:**

Please address the aformentioned weaknesses.

---

> ### Author Response · Authors · 2024-11-27
>
> We thank Reviewer 4o4x for taking the time to review our paper and for their constructive feedback. Please find below our point-to-point response:
>
> > **Computational Complexity Analysis**:
>
> Thank you for raising this important point. We have added a remark in the method section regarding computational complexity. Specifically, for exact Hessian matrix inverse computation, our method reduces the computational complexity from $\Omega\left(\left(\sum_{k=1}^K d_k + p\right)^w\right)$ to $\Omega\left(\sum_{k=1}^K d_k^w + p^w\right)$, where $w \approx 2.37$, $d_k$ are the dimension of task specific parameters for task $k$, and $p$ is the dimension of shared parameters. This reduction is achieved by decoupling shared and task-specific parameters in the optimization process. For a detailed discussion, please refer to the updated method section in the paper.
>
> **Empirical Approximation to the Hessian Inverse**:
>
> Motivated by your suggestion, we have expanded the discussion in the method section to address potential approximations for the Hessian inverse, focusing on two widely used approaches: EK-FAC and LiSSA. Our derivations provide insights into their applicability in MTL settings. The updated paper includes a detailed discussion, which we summarize below:
> - **EK-FAC**: This method approximates the Hessian using a blockwise diagonal matrix, which ignores off-diagonal interactions between shared and task-specific parameters. While computationally efficient, this approximation can lead to the loss of significant contributions when computing influence scores, particularly in soft parameter-sharing models where inter-task interactions play a critical role.
> - **LiSSA**: This method approximates the *inverse-Hessian-vector-product* using an iterative algorithm that supports mini-batch gradients. In MTL settings, however, the empirical Hessian for a data point has a unique structure due to parameter sharing, with non-zero entries restricted to specific sub-blocks. This structure often results in the mini-batch empirical Hessian being ill-posed, characterized by a high condition number, which poses challenges for achieving convergence and numerical stability. In this revision, we ran additional experiments to assess the applicability of LiSSA and added the results to the Appendix. Our empirical results suggest that, as the number of tasks increases, LiSSA requires progressively larger batch sizes to stabilize the stochastic approximation. This scaling significantly raises the computational costs for large-scale MTL problems. Adapting popular methods like LiSSA to address challenges arising from the unique Hessian structure in MTL settings requires nontrivial efforts and would be a valuable direction for future work.
>
> > **Experimental Scope**:
>
> We report the data selection results with different multitask architectures, namely CGC [5], MMoE[3] and DSelect-k [4]. The results demonstrate that, after applying data selection with MTIF, the test accuracy of most tasks shows significant improvement across all model architectures. These results have also been included in the Appendix Section C.3.3, and we list below for your reference.
>
> | Methods / Models | Task 1 | Task 2 | Task 3 | Task 4 | Task 5 | Task 6 | Task 7 | Task 8 | Task 9 | Average |
> |------------------|--------|--------|--------|--------|--------|--------|--------|--------|--------|---------|
> | CGC              | 0.863  | 0.772  | 0.878  | 0.734  | 0.920  | 0.939  | 0.834  | 0.900  | 0.949  | 0.866   |
> | CGC+DS           | 0.868  | 0.783  | 0.877  | 0.766  | 0.925  | 0.943  | 0.842  | 0.917  | 0.956  | 0.875   |
> | DSelect_k        | 0.855  | 0.786  | 0.862  | 0.758  | 0.927  | 0.947  | 0.850  | 0.913  | 0.950  | 0.872   |
> | DSelect_k+DS     | 0.868  | 0.787  | 0.867  | 0.775  | 0.933  | 0.951  | 0.856  | 0.924  | 0.954  | 0.880   |
> | HPS              | 0.859  | 0.815  | 0.896  | 0.791  | 0.934  | 0.951  | 0.872  | 0.919  | 0.958  | 0.888   |
> | HPS+DS           | 0.872  | 0.825  | 0.896  | 0.802  | 0.935  | 0.954  | 0.868  | 0.927  | 0.961  | 0.893   |
> | MMoE             | 0.843  | 0.793  | 0.881  | 0.740  | 0.917  | 0.944  | 0.842  | 0.899  | 0.956  | 0.868   |
> | MMoE+DS          | 0.868  | 0.793  | 0.887  | 0.766  | 0.929  | 0.949  | 0.867  | 0.926  | 0.959  | 0.883   |
>
> As for more complex datasets, we note that our experiments have included a fairly complex dataset, the CelebA dataset. The CelebA dataset was introduced at ICCV 2015 by [6], and comprises over 200,000 celebrity images, each annotated with 40 binary attributes, covering a wide range of facial features and expressions. Successful predictions on CelebA data require capturing the nuanced facial features in the image. Furthermore, CelebA has been widely used as a standard benchmark in the MTL literature [5], as it is natural to convert the annotated attributes into multiple tasks.

---

> ### Author Response · Authors · 2024-11-27
>
> > **Comparison with Baselines**:
>
> We have incorporated two gradient-based baselines, TAG [1] and Cosine Similarity [2], into our task-relatedness experiments for both linear regression and neural networks. These baselines are methods for measuring task relatedness in the MTL literature. Each baseline method provides a score of task relatedness for each pair of tasks. We evaluate these methods in terms of the correlation between their scores and the oracle task relatedness obtained from brute-force LOTO retraining as detailed in our paper.
>
>
> The results, as shown below, clearly demonstrate that our proposed MTIF method outperforms these baselines. Specifically, MTIF achieves consistently higher correlation coefficients with oracle influence estimates, underscoring its superior effectiveness in quantifying task-relatedness. We have included these new results in Appendix Section C.2, and we list below for your convenience.
>
> Results for synthetic dataset:
> | Method / Task | Task 1          | Task 2          | Task 3          | Task 4          | Task 5          |
> |---------------|-----------------|-----------------|-----------------|-----------------|-----------------|
> | Ours          | 0.84 ± 0.05    | 0.72 ± 0.05    | 0.74 ± 0.11    | 0.81 ± 0.05    | 0.71 ± 0.09    |
> | TAG           | 0.57 ± 0.03    | 0.63 ± 0.07    | 0.49 ± 0.11    | 0.56 ± 0.05    | 0.69 ± 0.04    |
> | Cosine        | 0.52 ± 0.04    | 0.48 ± 0.07    | 0.39 ± 0.12    | 0.47 ± 0.09    | 0.58 ± 0.06    |
>
> | Method / Task | Task 6          | Task 7          | Task 8          | Task 9          | Task 10         |
> |---------------|-----------------|-----------------|-----------------|-----------------|-----------------|
> | Ours          | 0.74 ± 0.04    | 0.74 ± 0.07    | 0.84 ± 0.03    | 0.74 ± 0.03    | 0.65 ± 0.07    |
> | TAG           | 0.55 ± 0.12    | 0.42 ± 0.06    | 0.44 ± 0.24    | 0.66 ± 0.08    | 0.61 ± 0.07    |
> | Cosine        | 0.47 ± 0.12    | 0.34 ± 0.05    | 0.40 ± 0.22    | 0.62 ± 0.09    | 0.51 ± 0.08    |
>
> Results for HAR dataset:
> | Method / Task | Task 1       | Task 2       | Task 3       | Task 4       | Task 5       | Task 6       |
> |---------------|--------------|--------------|--------------|--------------|--------------|--------------|
> | Ours          | 0.87 ± 0.02 | 0.90 ± 0.02 | 0.88 ± 0.01 | 0.91 ± 0.03 | 0.91 ± 0.01 | 0.90 ± 0.02 |
> | TAG           | 0.26 ± 0.13 | 0.42 ± 0.11 | 0.55 ± 0.09 | 0.22 ± 0.07 | 0.60 ± 0.07 | 0.55 ± 0.08 |
> | Cosine        | 0.31 ± 0.11 | 0.40 ± 0.11 | 0.57 ± 0.08 | 0.20 ± 0.09 | 0.61 ± 0.06 | 0.57 ± 0.08 |
>
> | Method / Task | Task 7       | Task 8       | Task 9       | Task 10      | Task 11      | Task 12      |
> |---------------|--------------|--------------|--------------|--------------|--------------|--------------|
> | Ours          | 0.90 ± 0.01 | 0.88 ± 0.02 | 0.92 ± 0.01 | 0.91 ± 0.02 | 0.89 ± 0.02 | 0.86 ± 0.01 |
> | TAG           | 0.49 ± 0.12 | 0.31 ± 0.12 | 0.24 ± 0.01 | 0.33 ± 0.02 | 0.43 ± 0.03 | 0.21 ± 0.02 |
> | Cosine        | 0.46 ± 0.11 | 0.31 ± 0.14 | 0.26 ± 0.03 | 0.34 ± 0.01 | 0.46 ± 0.04 | 0.18 ± 0.11 |
>
> | Method / Task | Task 13      | Task 14      | Task 15      | Task 16      | Task 17      | Task 18      |
> |---------------|--------------|--------------|--------------|--------------|--------------|--------------|
> | Ours          | 0.90 ± 0.02 | 0.93 ± 0.05 | 0.84 ± 0.01 | 0.87 ± 0.05 | 0.89 ± 0.02 | 0.82 ± 0.02 |
> | TAG           | 0.54 ± 0.03 | 0.57 ± 0.03 | 0.43 ± 0.02 | 0.48 ± 0.03 | 0.64 ± 0.05 | 0.44 ± 0.02 |
> | Cosine        | 0.53 ± 0.10 | 0.58 ± 0.10 | 0.48 ± 0.04 | 0.49 ± 0.11 | 0.66 ± 0.05 | 0.46 ± 0.07 |
>
> | Method / Task | Task 19      | Task 20      | Task 21      | Task 22      | Task 23      | Task 24      |
> |---------------|--------------|--------------|--------------|--------------|--------------|--------------|
> | Ours          | 0.85 ± 0.02 | 0.91 ± 0.02 | 0.93 ± 0.02 | 0.80 ± 0.01 | 0.80 ± 0.02 | 0.82 ± 0.05 |
> | TAG           | 0.44 ± 0.03 | 0.46 ± 0.02 | 0.84 ± 0.02 | 0.52 ± 0.07 | 0.13 ± 0.03 | 0.38 ± 0.07 |
> | Cosine        | 0.48 ± 0.05 | 0.47 ± 0.07 | 0.84 ± 0.10 | 0.53 ± 0.08 | 0.16 ± 0.12 | 0.45 ± 0.10 |
>
> | Method / Task | Task 25      | Task 26      | Task 27      | Task 28      | Task 29      | Task 30      |
> |---------------|--------------|--------------|--------------|--------------|--------------|--------------|
> | Ours          | 0.89 ± 0.02 | 0.81 ± 0.03 | 0.82 ± 0.03 | 0.89 ± 0.01 | 0.92 ± 0.03 | 0.86 ± 0.03 |
> | TAG           | 0.56 ± 0.04 | 0.14 ± 0.11 | 0.41 ± 0.10 | 0.14 ± 0.11 | 0.72 ± 0.04 | 0.41 ± 0.11 |
> | Cosine        | 0.60 ± 0.04 | 0.18 ± 0.12 | 0.46 ± 0.10 | 0.15 ± 0.10 | 0.74 ± 0.11 | 0.46 ± 0.10 |

---

> ### Author Response · Authors · 2024-11-27
>
> Results for CelebA:
> | Method / Task | Task 1       | Task 2       | Task 3       | Task 4       | Task 5       |
> |---------------|--------------|--------------|--------------|--------------|--------------|
> | Ours          | 0.23 ± 0.08 | 0.44 ± 0.19 | 0.25 ± 0.11 | 0.36 ± 0.12 | 0.17 ± 0.13 |
> | TAG           | -0.10 ± 0.13 | -0.10 ± 0.14 | 0.09 ± 0.06 | 0.40 ± 0.08 | 0.00 ± 0.12 |
> | Cosine        | 0.12 ± 0.18 | 0.08 ± 0.15 | 0.08 ± 0.07 | 0.37 ± 0.08 | -0.10 ± 0.13 |
>
> | Method / Task | Task 6       | Task 7       | Task 8       | Task 9       |
> |---------------|--------------|--------------|--------------|--------------|
> | Ours          | 0.35 ± 0.08 | 0.25 ± 0.07 | 0.11 ± 0.09 | 0.18 ± 0.12 |
> | TAG           | -0.42 ± 0.08 | -0.26 ± 0.17 | 0.06 ± 0.13 | 0.16 ± 0.16 |
> | Cosine        | -0.25 ± 0.12 | -0.25 ± 0.14 | -0.01 ± 0.16 | 0.05 ± 0.12 |
>
>
>
> We appreciate your insightful feedback, which has significantly strengthened our paper. Thank you again for your thoughtful comments!
>
> [1] Fifty, Chris, et al. "Efficiently identifying task groupings for multi-task learning." Advances in Neural Information Processing Systems 34 (2021): 27503-27516.
>
> [2] Azorin, Raphaël, et al. "" It's a Match!"--A Benchmark of Task Affinity Scores for Joint Learning." arXiv preprint arXiv:2301.02873 (2023).
>
> [3] Ma, Jiaqi, et al. "Modeling task relationships in multi-task learning with multi-gate mixture-of-experts." Proceedings of the 24th ACM SIGKDD international conference on knowledge discovery & data mining. 2018.
>
> [4] Hazimeh, Hussein, et al. "Dselect-k: Differentiable selection in the mixture of experts with applications to multi-task learning." Advances in Neural Information Processing Systems 34 (2021): 29335-29347.
>
> [5] Tang, Hongyan, et al. "Progressive layered extraction (ple): A novel multi-task learning (mtl) model for personalized recommendations." Proceedings of the 14th ACM Conference on Recommender Systems. 2020.
>
> [6] Ziwei Liu, Ping Luo, Xiaogang Wang, and Xiaoou Tang. Deep learning face attributes in the wild. In *Proceedings of the IEEE International Conference on Computer Vision (ICCV)*, December 2015
>
> [7] Alex Krizhevsky. Learning multiple layers of features from tiny images. 2009.

---

### Official Review · Reviewer_CA63 · 2024-11-09

**Soundness:** 2
**Presentation:** 2
**Contribution:** 2
**Rating:** 5
**Confidence:** 3

**Summary:**

The paper introduces a novel method called the MultiTask Influence Function (MTIF) for data attribution in multitask learning (MTL) settings, which extends data attribution from single-task learning to MTL, addressing both the opportunities and challenges that come with MTL.
﻿
1. **Novel Connection**: It establishes a new connection between data attribution and MTL, showing that data attribution can be used to efficiently measure task relatedness, a critical factor in MTL.
﻿
2. **MTIF Proposal**: The authors propose MTIF, a data attribution method designed for MTL. MTIF leverages the structure of MTL models to estimate the impact of removing data points or excluding tasks on the predictions of specific target tasks. It provides both data-level and task-level influence analysis.
﻿
3. **Efficiency and Scalability**: MTIF offers an efficient and scalable solution for data attribution in MTL by approximating leave-one-out and leave-one-task-out effects without the need for model retraining.
﻿
4. **Practical Usefulness**: MTIF can be used for practical applications such as data selection, which results in consistent performance improvements over baselines and helps mitigate negative transfer effects in MTL.
﻿
In summary, the paper presents an advancement in the field of multitask learning by introducing a method that enhances model interpretability and performance through efficient data attribution.

**Strengths:**

### Originality
﻿
1. **Innovative Approach to Data Attribution in MTL**: The paper introduces the MultiTask Influence Function (MTIF), which is a novel method for data attribution in multitask learning (MTL). This extends the application of data attribution beyond single-task learning, representing a creative advancement in the field.

2. **New Perspective on Task Relatedness**: By proposing a method to measure task relatedness in MTL, the paper offers a fresh metric for understanding task interactions, which is an original contribution to the understanding and optimization of MTL models.
﻿
### Quality
﻿
1. **Thorough Experimental Validation**: The paper provides a rigorous experimental framework, testing MTIF on both linear and neural network models, which speaks to the high quality of the research and its findings.
﻿
### Clarity
﻿
1. **Clear Problem Formulation**: The paper clearly defines the problem of data attribution in MTL, making it accessible to readers who may not be experts in the field.
﻿
2. **Detailed Methodological Explanation**: The step-by-step explanation of the MTIF methodology, including the mathematical derivations, enhances the clarity and understandability of the paper.

**Weaknesses:**

### Specificity in Application Domains
﻿
1. **Limited Domain Diversity**: The paper primarily focuses on synthetic and neural network models. While this provides a solid foundation, expanding the experiments to include a broader range of real-world datasets and application domains could strengthen the claims of generalizability.
﻿
### Depth of Negative Transfer Analysis
﻿
2. **Analysis of Negative Transfer**: While the paper mentions the mitigation of negative transfer, a more in-depth analysis of how MTIF specifically addresses and quantifies negative transfer effects could be beneficial.
﻿
### Comparative Analysis
﻿
3. **Lack of Comparative Analysis with Other Attribution Methods**: The paper could benefit from a comparative analysis with other existing data attribution methods in MTL to better highlight the advantages and potential limitations of MTIF.
﻿
### Robustness and Generalization
﻿
4. **Robustness Across Different Model Architectures**: More extensive testing of MTIF across different model architectures and complexities could provide a clearer picture of its robustness and generalization capabilities.

**Questions:**

1. Given that the paper primarily focuses on synthetic and neural network models, how might the findings differ if a broader range of real-world datasets and diverse application domains were included in the experiments? What steps could be taken to test the generalizability of the results in these different contexts?
﻿
2. The paper touches on the mitigation of negative transfer but lacks an in-depth analysis of MTIF's effectiveness in this regard. What specific methodologies or metrics could be employed to better quantify and analyze the negative transfer effects mitigated by MTIF? How would such an analysis strengthen the overall claims of the paper?

---

> ### Author Response · Authors · 2024-11-27
>
> We thank Reviewer CA63 for taking the time to review our paper and for their constructive feedback. Please find below our point-to-point response:
>
> > **Generalizability Across Different Datasets and Model Architectures**:
>
> We report the data selection accuracy with different multitask architectures, namely CGC [5], MMoE [3] and DSelect-k [4]. The results demonstrate that, after applying data selection with MTIF, the performance of most tasks shows significant improvement across all model architectures. These results have also been included in the Appendix Section C.3.3, and we list below for your reference.
>
> | Methods / Models | Task 1 | Task 2 | Task 3 | Task 4 | Task 5 | Task 6 | Task 7 | Task 8 | Task 9 | Average |
> |------------------|--------|--------|--------|--------|--------|--------|--------|--------|--------|---------|
> | CGC              | 0.863  | 0.772  | 0.878  | 0.734  | 0.920  | 0.939  | 0.834  | 0.900  | 0.949  | 0.866   |
> | CGC+DS           | 0.868  | 0.783  | 0.877  | 0.766  | 0.925  | 0.943  | 0.842  | 0.917  | 0.956  | 0.875   |
> | DSelect_k        | 0.855  | 0.786  | 0.862  | 0.758  | 0.927  | 0.947  | 0.850  | 0.913  | 0.950  | 0.872   |
> | DSelect_k+DS     | 0.868  | 0.787  | 0.867  | 0.775  | 0.933  | 0.951  | 0.856  | 0.924  | 0.954  | 0.880   |
> | HPS              | 0.859  | 0.815  | 0.896  | 0.791  | 0.934  | 0.951  | 0.872  | 0.919  | 0.958  | 0.888   |
> | HPS+DS           | 0.872  | 0.825  | 0.896  | 0.802  | 0.935  | 0.954  | 0.868  | 0.927  | 0.961  | 0.893   |
> | MMoE             | 0.843  | 0.793  | 0.881  | 0.740  | 0.917  | 0.944  | 0.842  | 0.899  | 0.956  | 0.868   |
> | MMoE+DS          | 0.868  | 0.793  | 0.887  | 0.766  | 0.929  | 0.949  | 0.867  | 0.926  | 0.959  | 0.883   |
>
> As for more complex datasets, we note that our experiments have included a fairly complex dataset, the CelebA dataset. The CelebA dataset was introduced at ICCV 2015 by [6], and comprises over 200,000 celebrity images, each annotated with 40 binary attributes, covering a wide range of facial features and expressions. Successful predictions on CelebA data require capturing the nuanced facial features in the image. Furthermore, CelebA has been widely used as a standard benchmark in the MTL literature [5], as it is natural to convert the annotated attributes into multiple tasks.
>
> > **Comparative Analysis with Other Attribution Methods**:
>
> To the best of our knowledge, there are currently no existing data attribution methods specifically tailored for multitask learning. Our work is the first to propose a framework that adapts data attribution methods to the MTL setting. In response to this feedback, we have added a comparison of task-relatedness measurements between our method and widely used gradient-based task-relatedness measures from the MTL literature in the updated version of the paper. Specifically, we have incorporated two gradient-based baselines, TAG [1] and Cosine Similarity [2], into our task-relatedness experiments for both linear regression and neural networks. Each baseline method provides a score of task relatedness for each pair of tasks. We evaluate these methods in terms of the correlation between their scores and the oracle task relatedness obtained from brute-force LOTO retraining as detailed in our paper.

---

> ### Author Response · Authors · 2024-11-27
>
> The results, as shown below, clearly demonstrate that our proposed MTIF method outperforms these baselines. Specifically, MTIF achieves consistently higher correlation coefficients with oracle influence estimates, underscoring its superior effectiveness in quantifying task-relatedness. We have included these new results in Appendix Section C.2, and we list below for your convenience.
>
> Results for synthetic dataset:
> | Method / Task | Task 1          | Task 2          | Task 3          | Task 4          | Task 5          |
> |---------------|-----------------|-----------------|-----------------|-----------------|-----------------|
> | Ours          | 0.84 ± 0.05    | 0.72 ± 0.05    | 0.74 ± 0.11    | 0.81 ± 0.05    | 0.71 ± 0.09    |
> | TAG           | 0.57 ± 0.03    | 0.63 ± 0.07    | 0.49 ± 0.11    | 0.56 ± 0.05    | 0.69 ± 0.04    |
> | Cosine        | 0.52 ± 0.04    | 0.48 ± 0.07    | 0.39 ± 0.12    | 0.47 ± 0.09    | 0.58 ± 0.06    |
>
> | Method / Task | Task 6          | Task 7          | Task 8          | Task 9          | Task 10         |
> |---------------|-----------------|-----------------|-----------------|-----------------|-----------------|
> | Ours          | 0.74 ± 0.04    | 0.74 ± 0.07    | 0.84 ± 0.03    | 0.74 ± 0.03    | 0.65 ± 0.07    |
> | TAG           | 0.55 ± 0.12    | 0.42 ± 0.06    | 0.44 ± 0.24    | 0.66 ± 0.08    | 0.61 ± 0.07    |
> | Cosine        | 0.47 ± 0.12    | 0.34 ± 0.05    | 0.40 ± 0.22    | 0.62 ± 0.09    | 0.51 ± 0.08    |
>
> Results for HAR dataset:
> | Method / Task | Task 1       | Task 2       | Task 3       | Task 4       | Task 5       | Task 6       |
> |---------------|--------------|--------------|--------------|--------------|--------------|--------------|
> | Ours          | 0.87 ± 0.02 | 0.90 ± 0.02 | 0.88 ± 0.01 | 0.91 ± 0.03 | 0.91 ± 0.01 | 0.90 ± 0.02 |
> | TAG           | 0.26 ± 0.13 | 0.42 ± 0.11 | 0.55 ± 0.09 | 0.22 ± 0.07 | 0.60 ± 0.07 | 0.55 ± 0.08 |
> | Cosine        | 0.31 ± 0.11 | 0.40 ± 0.11 | 0.57 ± 0.08 | 0.20 ± 0.09 | 0.61 ± 0.06 | 0.57 ± 0.08 |
>
> | Method / Task | Task 7       | Task 8       | Task 9       | Task 10      | Task 11      | Task 12      |
> |---------------|--------------|--------------|--------------|--------------|--------------|--------------|
> | Ours          | 0.90 ± 0.01 | 0.88 ± 0.02 | 0.92 ± 0.01 | 0.91 ± 0.02 | 0.89 ± 0.02 | 0.86 ± 0.01 |
> | TAG           | 0.49 ± 0.12 | 0.31 ± 0.12 | 0.24 ± 0.01 | 0.33 ± 0.02 | 0.43 ± 0.03 | 0.21 ± 0.02 |
> | Cosine        | 0.46 ± 0.11 | 0.31 ± 0.14 | 0.26 ± 0.03 | 0.34 ± 0.01 | 0.46 ± 0.04 | 0.18 ± 0.11 |
>
> | Method / Task | Task 13      | Task 14      | Task 15      | Task 16      | Task 17      | Task 18      |
> |---------------|--------------|--------------|--------------|--------------|--------------|--------------|
> | Ours          | 0.90 ± 0.02 | 0.93 ± 0.05 | 0.84 ± 0.01 | 0.87 ± 0.05 | 0.89 ± 0.02 | 0.82 ± 0.02 |
> | TAG           | 0.54 ± 0.03 | 0.57 ± 0.03 | 0.43 ± 0.02 | 0.48 ± 0.03 | 0.64 ± 0.05 | 0.44 ± 0.02 |
> | Cosine        | 0.53 ± 0.10 | 0.58 ± 0.10 | 0.48 ± 0.04 | 0.49 ± 0.11 | 0.66 ± 0.05 | 0.46 ± 0.07 |
>
> | Method / Task | Task 19      | Task 20      | Task 21      | Task 22      | Task 23      | Task 24      |
> |---------------|--------------|--------------|--------------|--------------|--------------|--------------|
> | Ours          | 0.85 ± 0.02 | 0.91 ± 0.02 | 0.93 ± 0.02 | 0.80 ± 0.01 | 0.80 ± 0.02 | 0.82 ± 0.05 |
> | TAG           | 0.44 ± 0.03 | 0.46 ± 0.02 | 0.84 ± 0.02 | 0.52 ± 0.07 | 0.13 ± 0.03 | 0.38 ± 0.07 |
> | Cosine        | 0.48 ± 0.05 | 0.47 ± 0.07 | 0.84 ± 0.10 | 0.53 ± 0.08 | 0.16 ± 0.12 | 0.45 ± 0.10 |
>
> | Method / Task | Task 25      | Task 26      | Task 27      | Task 28      | Task 29      | Task 30      |
> |---------------|--------------|--------------|--------------|--------------|--------------|--------------|
> | Ours          | 0.89 ± 0.02 | 0.81 ± 0.03 | 0.82 ± 0.03 | 0.89 ± 0.01 | 0.92 ± 0.03 | 0.86 ± 0.03 |
> | TAG           | 0.56 ± 0.04 | 0.14 ± 0.11 | 0.41 ± 0.10 | 0.14 ± 0.11 | 0.72 ± 0.04 | 0.41 ± 0.11 |
> | Cosine        | 0.60 ± 0.04 | 0.18 ± 0.12 | 0.46 ± 0.10 | 0.15 ± 0.10 | 0.74 ± 0.11 | 0.46 ± 0.10 |
>
>
> Results for CelebA dataset:
>
> | Method / Task | Task 1       | Task 2       | Task 3       | Task 4       | Task 5       |
> |---------------|--------------|--------------|--------------|--------------|--------------|
> | Ours          | 0.23 ± 0.08 | 0.44 ± 0.19 | 0.25 ± 0.11 | 0.36 ± 0.12 | 0.17 ± 0.13 |
> | TAG           | -0.10 ± 0.13 | -0.10 ± 0.14 | 0.09 ± 0.06 | 0.40 ± 0.08 | 0.00 ± 0.12 |
> | Cosine        | 0.12 ± 0.18 | 0.08 ± 0.15 | 0.08 ± 0.07 | 0.37 ± 0.08 | -0.10 ± 0.13 |
>
> | Method / Task | Task 6       | Task 7       | Task 8       | Task 9       |
> |---------------|--------------|--------------|--------------|--------------|
> | Ours          | 0.35 ± 0.08 | 0.25 ± 0.07 | 0.11 ± 0.09 | 0.18 ± 0.12 |
> | TAG           | -0.42 ± 0.08 | -0.26 ± 0.17 | 0.06 ± 0.13 | 0.16 ± 0.16 |
> | Cosine        | -0.25 ± 0.12 | -0.25 ± 0.14 | -0.01 ± 0.16 | 0.05 ± 0.12 |

---

> ### Author Response · Authors · 2024-11-27
>
> > **Analysis of Negative Transfer**:
>
> MTIF is designed to estimate the influence of a data point or source task on the performance of a target task. The calculated influence score provides a clear and interpretable measure of transfer effects: positive influence scores indicate potential positive transfer, while negative influence scores highlight potential negative transfer. In our experiments, we leverage this property for data selection by filtering out data points with low influence scores to mitigate negative transfer. Our results show that this data selection strategy, guided by our influence scores, effectively improves the performance of MTL algorithms. These findings underscore the utility of MTIF in addressing negative transfer and enhancing multitask learning outcomes.
>
> We appreciate your insightful feedback, which has significantly strengthened our paper. Thank you again for your thoughtful comments!
>
> [1] Fifty, Chris, et al. "Efficiently identifying task groupings for multi-task learning." Advances in Neural Information Processing Systems 34 (2021): 27503-27516.
>
> [2] Azorin, Raphaël, et al. "" It's a Match!"--A Benchmark of Task Affinity Scores for Joint Learning." arXiv preprint arXiv:2301.02873 (2023).
>
> [3] Ma, Jiaqi, et al. "Modeling task relationships in multi-task learning with multi-gate mixture-of-experts." Proceedings of the 24th ACM SIGKDD international conference on knowledge discovery & data mining. 2018.
>
> [4] Hazimeh, Hussein, et al. "Dselect-k: Differentiable selection in the mixture of experts with applications to multi-task learning." Advances in Neural Information Processing Systems 34 (2021): 29335-29347.
>
> [5] Tang, Hongyan, et al. "Progressive layered extraction (ple): A novel multi-task learning (mtl) model for personalized recommendations." Proceedings of the 14th ACM Conference on Recommender Systems. 2020.
>
> [6] Ziwei Liu, Ping Luo, Xiaogang Wang, and Xiaoou Tang. Deep learning face attributes in the wild. In *Proceedings of the IEEE International Conference on Computer Vision (ICCV)*, December 2015

---

### Author Response · Authors · 2024-11-27
**Message To All Reviewers**

We sincerely thank all the reviewers for their thoughtful feedback and valuable suggestions. We apologize for the delay in our response, as we added a series of additional analyses and experiments to comprehensively address the reviewers’ comments. Specifically, we have made the following major updates:

**Discussion of Computational Complexity**: We have added a detailed analysis of the computational complexity gains achieved by our method for computing the exact Hessian inverse, highlighting the efficiency improvements over naive approaches.

**Analysis of Approximation Algorithms**: We have expanded the discussion to include an analysis of potential efficient approximation algorithms, such as EK-FAC and LiSSA, and evaluated their applicability and limitations in the MTL setting.

**Additional Experimental Baselines**: We have incorporated new experimental baselines for comparison to provide a more comprehensive evaluation of our method. These include TAG [1] and Cosine Similarity [2] from the conventional MTL literature, which serve as baselines for measuring task relatedness.

We have addressed all the comments in the detailed individual response to each reviewer, as well as updated our paper draft to reflect the changes. Here we would like to highlight a few key points in our response.

Firstly, to the best of our knowledge, this is the first paper to introduce data attribution methods in multitask learning (MTL) settings. Reviewers QvTU and MsCP raised concerns about the distinctions between our proposed method and single-task learning (STL)-based influence function methods. We acknowledge that our proposed MTL shares conceptual similarities with STL-based influence functions. However, our contributions extend beyond existing work in the following key dimensions:
1. Adapting influence functions to the MTL setting requires a novel framework that accounts for the unique parameter and model evaluation structures inherent to MTL.
2. Our method introduces a natural mechanism to estimate task-relatedness and mitigate negative transfer—two critical challenges in MTL. These aspects are particularly relevant and impactful within the MTL literature, as they address fundamental issues in multitask optimization and learning.
3. Our derivations provide new insights into the applicability and limitations of popular STL-based Hessian inverse approximation methods, such as EK-FAC and LiSSA, when applied to MTL. This bridges a gap in the literature and opens avenues for further research on scalable approximations in multitask settings.

Secondly, we have conducted extensive additional experiments, including the following key ones:
1) We have incorporated two gradient-based baselines from conventional MTL literature, TAG [1] and Cosine Similarity [2], for measuring task relatedness. The results, provided in Appendix Section C.2, clearly demonstrate that our proposed MTIF method outperforms these baselines. Specifically, MTIF achieves consistently higher correlation coefficients with the oracle LOTO influences, underscoring its superior effectiveness in quantifying task-relatedness.

2) We have experimented with several additional MTL model architectures that have been widely cited in MTL literature. We demonstrate that combining data selection enabled by MTIF with these model architectures consistently improves their MTL performance. This further confirms that the proposed MTIF enables a novel method to improve MTL complementary to most existing methods in the MTL literature.

We hope this clarification adequately addresses the reviewers’ concerns and highlights the distinct contributions of our work. Thank you again for your insightful feedback.

[1] Fifty, Chris, et al. "Efficiently identifying task groupings for multi-task learning." Advances in Neural Information Processing Systems 34 (2021): 27503-27516.

[2] Azorin, Raphaël, et al. "" It's a Match!"--A Benchmark of Task Affinity Scores for Joint Learning." arXiv preprint arXiv:2301.02873 (2023).

---

### Author Response · Authors · 2024-11-30
**Discussion Appreciated**

Dear Reviewers,

Thank you once again for your valuable and constructive feedback. In our response, we have significantly revised our paper, incorporating substantial additional experiments and analyses. We have also carefully addressed each review comment for every reviewer in the individual responses.

As the discussion period comes to a close, we would greatly appreciate any further opportunities to engage with you in continued discussion. Thank you for your time and consideration.

Best regards,

The authors of submission 8237: Data Attribution for Multitask Learning

---

### Meta-Review · Area_Chair_S5LX · 2024-12-19

**Metareview:**

This paper examines Data Attribution methods for Multi-Task Learning (MTL) through the lens of the influence function. Following the rebuttal, it received an overall borderline score. I reviewed both the discussions and the paper myself. Three main issues still need resolution:

- **Effectiveness on Larger Models**: Our reviewer has noted that this paper focuses solely on traditional deep learning models, whereas modern MTL approaches are based on foundation models or LLMs. It remains uncertain if the hessian-based computation can be applied to these models.

- **Validation on Larger Datasets**: The evaluation is limited to a few real-world datasets, with a few tasks considered. Thus, whether the proposed method can effectively scale to more extensive task sets is unclear.

-  **Analysis of Negative Transfer**:  While the paper mentions mitigating negative transfer, a more in-depth analysis of how MTIF specifically addresses and quantifies negative transfer is missing.

Overall, I recommend rejecting this paper. The authors should enhance their experiments significantly before resubmitting in the next round.

**Additional Comments On Reviewer Discussion:**

`4o4x` and `yCgf` are weakly positive after rebuttal. I agree that this paper proposes a novel way to attack MTL through data attribution.

However, my decision to reject this paper is out of the following concerns:

- **Effectiveness on Larger Models** (`CA63`): Our reviewer noted that this paper focuses solely on traditional deep learning models, whereas modern MTL approaches are based on foundation or LLMs. It remains uncertain if the hessian-based computation can be applied to these models.

- **Validation on Larger Datasets** (`edvo`): The evaluation is limited to a few real-world datasets, with a few tasks considered (CelebA should be viewed as a small dataset nowadays). Thus, whether the proposed method can effectively scale to more extensive task sets is unclear.

- **Analysis of Negative Transfer**(`CA63`):   While the paper mentions mitigating negative transfer, a more in-depth analysis of how MTIF specifically addresses and quantifies negative transfer is missing.

---

### Decision · Program_Chairs · 2025-01-22

Reject